# A Direct Approximation of AIXI Using Logical State Abstractions

**Samuel Yang-Zhao**[*][†]
Australian National University
Canberra ACT 2601
samuel.yang-zhao@anu.edu.au

**Tianyu Wang**[*]
Australian National University
Canberra ACT 2601
tianyu.wang2@anu.edu.au

**Kee Siong Ng**
Australian National University
Canberra ACT 2601
keesiong.ng@anu.edu.au

## Abstract

We propose a practical integration of logical state abstraction with AIXI, a Bayesian optimality notion for reinforcement learning agents, to significantly expand the model class that AIXI agents can be approximated over to complex history-dependent and structured environments. The state representation and reasoning framework is based on higher-order logic, which can be used to define and enumerate complex features on non-Markovian and structured environments. We address the problem of selecting the right subset of features to form state abstractions by adapting the $\Phi$-MDP optimisation criterion from state abstraction theory. Exact Bayesian model learning is then achieved using a suitable generalisation of Context Tree Weighting over abstract state sequences. The resultant architecture can be integrated with different planning algorithms. Experimental results on controlling epidemics on large-scale contact networks validates the agent's performance.

## 1 Introduction

The AIXI agent [1] is a mathematical solution to the general reinforcement learning problem that combines Solomonoff induction [2] with sequential decision theory. At time $t$, after observing the observation-reward sequence $or_{1:t-1}$ and performing previous actions $a_{1:t-1}$ the AIXI agent chooses action $a_t$ given by

$$a_t = \arg \max_{a_t} \sum_{or_t} \ldots \max_{a_{t+m}} \sum_{or_{t+m}} [r_t + \ldots + r_{t+m}] \sum_{q:U(q,a_{1:t+m})=or_{1:t+m}} 2^{-l(q)}, \qquad (1)$$

where $U$ is a universal Turing machine, $U(q, a_{1:n})$ is the output of $U$ given program $q$ and input $a_{1:n}$, $m \in \mathbb{N}$ is the lookahead horizon, and $l(q)$ is the length of $q$ in bits. [1] shows that AIXI's environment model converges rapidly to the true environment and its policy is pareto optimal and self-optimising.

While mathematically elegant, a fundamental issue with AIXI is that it is only asymptotically computable. The question of whether a good practical approximation of AIXI can be constructed remains an open problem twenty years after AIXI was first introduced [3]. The most direct approximation of AIXI given so far is [4], in which the Bayesian mixture over all Turing machines is approximated over $n$-Markov environments using Context Tree Weighting [5] and the expectimax search is approximated

---

[*]Equal contribution.
[†]Corresponding author.

using $\rho$UCT, a Monte Carlo planning algorithm [6]. This paper extends [4] in several ways. To move beyond $n$-Markov environments and make it easy to incorporate background knowledge, we adopt a formal knowledge representation and reasoning framework based on higher-order logic to allow complex features to be defined on non-Markovian, structured environments. The Feature Reinforcement Learning framework [7] is adapted to define an optimisation problem for selecting the best state abstractions and we present an algorithm based on Binary Decision Diagrams to constrain the set of solutions. Generalizing binarised CTW [8] to incorporate complex features in its context then allows a Bayesian mixture to be computed over a much larger model class. Combined with $\rho$UCT, the resulting agent is the richest, direct approximation of AIXI known to date. The practicality of the resultant agent is shown on epidemic control on non-trivial contact networks.

## 2 Background and Related Work

### 2.1 The General Reinforcement Learning Setting

We consider finite action, observation and reward spaces denoted by $\mathcal{A}, \mathcal{O}, \mathcal{R}$ respectively. The agent interacts with the environment in cycles: at any time, the agent chooses an action from $\mathcal{A}$ and the environment returns an observation and reward from $\mathcal{O}$ and $\mathcal{R}$. Frequently we will be considering observations and rewards together, and will denote $x \in \mathcal{O} \times \mathcal{R}$ as a percept $x$ from the percept space $\mathcal{O} \times \mathcal{R}$. We will denote a string $x_1 x_2 \ldots x_n$ of length $n$ by $x_{1:n}$ and its length $n-1$ prefix as $x_{<n}$. An action, observation and reward from the same time step will be denoted $aor_t$. After interacting for $n$ cycles, the interaction string $a_1 o_1 r_1 \ldots a_n o_n r_n$ (denoted $aor_{1:n}$ from here on) is generated. A history $h$ is an element of the history space $\mathcal{H} := (\mathcal{A} \times \mathcal{O} \times \mathcal{R})^*$. An environment $\rho$ is a sequence of probability distributions $\{\rho_0, \rho_1, \rho_2, \ldots\}$, where $\rho_n : \mathcal{A}^n \to \mathcal{D}((\mathcal{O} \times \mathcal{R})^n)$, that satisfies $\forall a_{1:n} \, \forall or_{<n} \, \rho_{n-1}(or_{<n}|a_{<n}) = \sum_{or \in \mathcal{O} \times \mathcal{R}} \rho_n(or_{1:n}|a_{1:n})$. We will drop the subscript on $\rho_n$ when the context is clear. The predictive probability of the next percept given history and a current action is given by $\rho(or_n|aor_{<n}, a_n) = \rho(or_n|h_{n-1} a_n) := \frac{\rho(or_{1:n}|a_{1:n})}{\rho(or_{<n}|a_{<n})}$ for all $aor_{1:n}$ such that $\rho(or_{<n}|a_{<n}) > 0$.

The general reinforcement learning problem is for the agent to learn a *policy* $\pi : \mathcal{H} \to \mathcal{D}(A)$ mapping histories to a distribution on possible actions that will allow it to maximise its future expected reward. In this paper, we consider the future expected reward up to a finite horizon $m \in \mathbb{N}$. Given a history $h$ and a policy $\pi$ the value function with respect to the environment $\rho$ is given by $V_\rho^\pi(h_t) := \mathbb{E}_\rho^\pi \left[ \sum_{i=t+1}^{t+m} R_i | h_t \right]$, where $R_i$ denotes the random variable distributed according to $\rho$ for the reward at time $i$. The action value function is defined similarly as $V_\rho^\pi(h_t, a_{t+1}) := \mathbb{E}_\rho^\pi \left[ \sum_{i=t+1}^{t+m} R_i | h_t, a_{t+1} \right]$. The agent's goal is to learn the optimal policy $\pi^*$, which is the policy that results in the value function with the maximum reward for any given history.

### 2.2 AIXI agent and its Monte-Carlo approximation

The following equation can be shown to be formally equivalent to Equation (1). At time $t$, the AIXI agent chooses action $a_t^*$ according to

$$a_t^* = \arg \max_{a_t} \sum_{or_t} \ldots \max_{a_{t+m}} \sum_{or_{t+m}} [r_t + \ldots r_{t+m}] \sum_{\rho \in \mathcal{M}_U} 2^{-K(\rho)} \rho(or_{1:t+m}|a_{1:t+m}), \quad (2)$$

where $m \in \mathbb{N}$ is a finite lookahead horizon, $\mathcal{M}_U$ is the set of all enumerable chronological semimeasures [1], $\rho(or_{1:t+m}|a_{1:t+m})$ is the probability of observing $or_{1:t+m}$ given the action sequence $a_{1:t+m}$, and $K(\rho)$ denotes the Kolmogorov complexity [9] of $\rho$.

The Bayesian mixture $\xi_U := \sum_{\rho \in \mathcal{M}_U} 2^{-K(\rho)} \rho(or_{1:t}|a_{1:t})$ in (2) is a mixture environment model, with $2^{-K(\rho)}$ as the prior for $\rho$. A mixture environment model enjoys the property that it converges rapidly to the true environment if there exists a 'good' model in the model class.

**Theorem 1.** *[1] Let $\mu$ be the true environment and $\xi$ be the mixture environment model over a model class $\mathcal{M}$. For all $n \in \mathbb{N}$ and for all $a_{1:n}$,*

$$\sum_{k=1}^{n} \sum_{or_{1:k}} \mu(or_{<k}|a_{<k}) \left( \mu(or_k|aor_{<k} a_k) - \xi(or_k|aor_{<k} a_k) \right)^2 \leq \min_{\rho \in \mathcal{M}} \left\{ \ln \frac{1}{w_0^\rho} + D_n(\mu||\rho) \right\} \quad (3)$$

where $D_n(\mu||\rho)$ is the KL divergence of $\mu(\cdot|a_{1:n})$ and $\rho(\cdot|a_{1:n})$.

To see the rapid convergence of $\xi$ to $\mu$, take the limit $n \to \infty$ on the l.h.s of (3) and observe that in the case where $\min_{\rho \in \mathcal{M}} \sup_n D_n(\mu||\rho)$ is bounded, the l.h.s. can only be finite if $\xi(or_k|aor_{<k}a_k)$ converges sufficiently fast to $\mu(or_k|aor_{<k}a_k)$. From Theorem 1, it can be shown that AIXI's environment mixture model $\xi_U$ also enjoys this property with any computable environment.

The first practical approximation of AIXI is given in [4], in which the expectimax search is approximated using Monte Carlo Tree Search [6] and Bayesian mixture learning is performed using Context Tree Weighting (CTW) [5]. There is a body of work on extending the Bayesian mixture learning to larger classes of history-based models [10, 11, 12, 13]. In this paper, we generalise the Bayesian mixture learning in an orthogonal direction by adapting binarised CTW [8] to run on sequences defined by logical state abstractions, thus significantly improving the ease with which symbolic and statistical background knowledge can be incorporated into the AIXI framework.

### 2.3 State Abstraction and Feature Reinforcement Learning

In the general reinforcement learning setting, the state abstraction framework is concerned with finding a function $\phi : \mathcal{H} \to \mathcal{S}_\phi$ that maps every history string to an element of an abstract state space $\mathcal{S}_\phi$. Given a state abstraction $\phi$, the abstract environment is a process that generates a state $s'$ and a reward $r$ given a history and action according to the distribution $\rho_\phi(s', r|h, a) := \sum_{o:\phi(haor)=s'} \rho(or|ha)$. The interaction sequence of the original process converts to a state-action-reward sequence under $\phi$.

A key reason for considering abstractions is that the space $\mathcal{H}$ is too large and the underlying process may be highly irregular and difficult to learn on. The aim is thus to select a mapping $\phi$ such that the induced process can facilitate learning without severely compromising performance. Theoretical approaches to this question have typically focused on providing conditions that minimise the error in the action-value function between the abstract and original process [14, 15]. The $\Phi$MDP framework instead provides an optimisation criteria for ranking candidate mappings $\phi$ based on how well the state-action-reward sequence generated by $\phi$ can be modelled as an MDP whilst being predictive of the rewards [7]. A good $\phi$ results in a Markovian model where the state is a sufficient statistic of the history and thus facilitates reinforcement learning.

**Optimisation Criteria for Selecting Mappings.** The $\Phi$MDP framework ranks candidate state abstractions based on an optimisation criteria defined over the code length of the state-reward sequence. Let $CL(x_{1:n})$ denote the code length of a sequence $x_{1:n}$ and $CL(x_{1:n}|a_{1:n})$ denote the code length of sequence $x_{1:n}$ conditioned on $a_{1:n}$. At time $n$, the $Cost_M$ objective is defined as

$$Cost_M(n, \phi) := CL(s_{1:n}|a_{1:n}) + CL(r_{1:n}|s_{1:n}, a_{1:n}) + CL(\phi) \qquad (4)$$

where $\phi(h_t) = s_t$ and $h_t = aor_{1:t}$. For more details on how MDP sequences can be coded see [7].

The objective is then to look for $\phi_n^{best} := \arg\min_\phi Cost_M(n, \phi)$. Note that $CL(\phi)$ is a term that imposes a Occam's razor-like penalty on the candidate mapping; if different candidate abstractions result in the same $Cost_M$ value, the simplest model is preferred.

For the agent to achieve high expected long-term reward, it is imperative that a state abstraction produces a model that can predict the reward sequence well. However, the equal weighting given to the code lengths of the state sequence in $Cost_M$ can be suboptimal; in fact a simple environment can be constructed where $Cost_M$ gives unintuitive results (see Appendix F). Given that $CL(s_{1:n}|a_{1:n})$ is optimised independent of any reward signal it does not inherently provide any information about the long-term reward. Thus, it is best thought of as a regularisation term to make the resulting MDP easier to learn. We drop the code length of the state sequence from the optimisation criteria to focus purely on reward predictability and hence consider $Cost_M^0(n, \phi) = CL(r_{1:n}|s_{1:n}, a_{1:n}) + CL(\phi)$.

### 2.4 Knowledge Representation and Reasoning Formalism

The formal agent knowledge representation and reasoning language is the one described in [16, 17]. The syntax of the language are the terms of the $\lambda$-calculus [18], extended with type polymorphism to increase its expressiveness. The inference engine has two interacting components: an equational-reasoning engine and a tableau theorem prover [19]. Thus, a predicate evaluation at the internal node of a Context Tree takes the general form $\Gamma \vdash p(h)$, where $\Gamma$ is a theory consisting of equations

and implications that reflects the agent's current belief base, $h$ is the history of agent-environment interacts, and $p$ is a predicate mapping histories to boolean values. There is also a predicate rewrite system for defining and enumerating a set of predicates.

Suitable alternatives to our logical formalism are studied in the Statistical Relational Artificial Intelligence literature [20, 21], including a few that caters specifically for relational reinforcement learning [22, 23] and Symbolic MDP/POMDPs [24, 25]. Our choice is informed by the standard arguments given in [26, 27] and the practical convenience of working with (the functional subset of) a language like Python. Our thesis that formal logic can be used to incorporate domain knowledge and broaden the model class of AIXI approximations remains unchanged with an alternate formal logic.

# 3 The $\Phi$-AIXI-CTW Agent

We now present the $\Phi$-AIXI-CTW agent. The actions and rewards are considered in their binary representation. The agent has access to a large set $\mathcal{P}$ of predicates on histories, and each subset of $\mathcal{P}$ constitutes a candidate state abstraction. We show that a $\Phi$-BCTW data structure performs a Bayesian mixture over all abstractions defined by $\mathcal{P}$ and will optimise $Cost_M^0$. However, this is computationally infeasible in practice if $\mathcal{P}$ is large. We first reduce this computational burden by performing feature selection to shrink the set of predicates for $\Phi$-BCTW. Finally, the agent selects actions according to the $\rho$UCT algorithm.

## 3.1 Shrinking the Model Class via Feature Selection

Our method for selecting candidate state abstractions is centred around the detection of predicate features that result in low entropy conditional reward distributions since lower entropy results in smaller code lengths and lower $Cost_M^0$. A simple heuristic that we employ is to look for candidate state abstractions that contain states that maximise the probability of receiving a given reward. We call such abstractions sharp mappings.

**Definition 1.** Let $p_\phi(r) := \sum_{s,a} p_\phi(r|s,a)$, $p_\phi(r^c|s,a) := \sum_{r' \neq r \in \mathcal{R}} p_\phi(r'|s,a)$ and $p_\phi(r^c) := \sum_{r' \neq r \in \mathcal{R}} p_\phi(r')$ for all $r \in \mathcal{R}$. For a $D > p_\phi(r)/p_\phi(r^c)$, an MDP mapping $\phi$ is $D$-sharp if for all $r \in \mathcal{R}$, there exists $s \in S_\phi$, $a \in \mathcal{A}$ such that $\frac{p_\phi(r|s,a)}{p_\phi(r^c|s,a)} \geq D$.

Sharp mappings are defined in this manner to help select candidate state abstractions that are predictive of all rewards. It is often the case in reinforcement learning environments that informative reward signals are sparse during learning. Treating rewards as class labels, the problem of predicting rewards can be viewed as an unbalanced data classification task. If the unbalanced nature of the data is not taken into account, feature selection methods will tend to ignore features that may be predictive of rewards that occur infrequently but are important for performance. Thus, we adjust for this fact by ensuring that the threshold chosen for each reward is bounded from below by $\frac{p_\phi(r)}{p_\phi(r^c)}$; if a reward $r$ occurs infrequently relative to the frequency of all other rewards, then the lower bound on $D$ will be small. Our approach is to select predicate features that lead to sharp mappings.

**Feature Selection Using Decision Boundaries.** Using the sharpness threshold, we frame the problem as a binary classification task. For each reward $r \in \mathcal{R}$, the sharpness of the reward distribution at a state-action pair determines the binary class by a decision rule defined by $F_{\phi,D}^r(s,a) := \mathbb{I}\left[\frac{p_\phi(r|s,a)}{p_\phi(r^c|s,a)} > D\right]$ with indicator function $\mathbb{I}$. Feature selection on the binary classification task can then be performed by eliminating conditionally redundant features, defined next.

**Definition 2.** Let $\phi_1 = \{p_1, \ldots, p_n\}$ and $\phi_2 = \phi_1 \setminus \{p_j\}$ for $j \in \{1, \ldots, n\}$ where $p_i$ is a predicate function. If $F_{\phi_1,D}^r(s,a) = F_{\phi_2,D}^r(s',a)$ for all $s \in \mathcal{S}_{\phi_1}$ and $s' = s_{1:j-1}s_{j+1:n}$, then $p_j$ is a conditionally redundant feature.

A conditionally redundant feature does not affect the output of the decision rule given the feature ordering and thus does not contribute to discriminating between different classes. As this property only depends upon the feature ordering upon construction, removing a conditionally redundant feature does not change whether the remaining features are redundant or not. To select candidate predicate features, we set up a decision rule for each reward $r \in \mathcal{R}$ and eliminate the predicate features that are redundant across all rewards. In practice, the probabilities used by the decision rules are estimated as frequency counts.

To remove conditionally redundant features, we use a Binary Decision Diagram (BDD) data structure [28] to represent each decision rule. A reduction procedure on the BDD representation then guarantees that only informative features remain.

**Theorem 2.** *Let $F_{\phi,D}^r : \mathcal{S}_\phi \times \mathcal{A} \to \{0,1\}$ be the decision rule for reward $r \in \mathcal{R}$. Then the reduced BDD representation of $F_{\phi,D}^r$ contains only conditionally informative features.*

The proof is given in Appendix E. In practice there exists simple procedures to reduce a BDD. For each reward, we determine a decision rule and perform BDD reduction to get a set of informative predicates. The union of the sets of selected predicates is then chosen.

One issue with BDD reduction is that the given variable ordering can impact whether a feature is conditionally redundant or not; a sub-optimal variable ordering will keep more features than necessary. We alleviate this issue by performing BDD reduction on multiple randomly selected subspaces of the feature space, corresponding to randomly selected subsets of predicates. We name our approach Random Forest-BDD (RF-BDD). For each selected subset of predicates $\mathcal{P}_i$, let $\phi_i$ be the corresponding MDP mapping. We perform BDD reduction on the decision rule constructed from $\mathcal{P}_i$ given by $F_{\mathcal{P}_i}^r(s,a) := \mathbb{I}\left[\frac{p_{\phi_i}(r|s,a)}{p_{\phi_i}(r^c|s,a)} > D\right]$. Once BDD reduction has been performed for all chosen subsets, a choice has to be made on which predicates to keep. We employ a voting scheme where a predicate receives a vote if it is kept by a BDD reduction. The votes are then normalised by the number of times each predicate was selected. The predicates that have a retention rate above a certain threshold will then be kept. The kept predicates for each percept are then combined into a single set.

We emphasise that selecting features based on a sharpness threshold does not directly optimise the $Cost_M^0$ criteria but rather constrains the set of possible solutions by their reward predictability. Thus the set of predicates selected by the RF-BDD procedure above, when input into $\Phi$-BCTW, lead to a model that approximately minimises $Cost_M^0$ over the initial set of predicates.

### 3.2 $\Phi$-Binarized Context Tree Weighting

The binarized context-tree weighting (BCTW) algorithm is a universal data compression scheme that extends the context-tree weighting algorithm to non-binary alphabets. The context-tree weighting algorithm is a data structure that computes an online Bayesian mixture model over prediction suffix trees [5]. The BCTW approach is to binarize a target symbol and predict each bit using a separate context tree. We present the $\Phi$-BCTW algorithm which modifies BCTW by allowing nodes to correspond to arbitrary predicates on histories. We show that this extension allows $\Phi$-BCTW to compute a Bayesian mixture model over environments that can be modelled by a set of predicate functions and will optimise $Cost_M^0$.

The $\Phi$-BCTW algorithm modifies the BCTW algorithm to allow it to model the state and reward transitions of an abstract environment defined under a set of predicate functions. Suppose we are given a set of predicates that defines the state space. The $\Phi$-BCTW models the abstract environment by assuming it has MDP structure and thus conditions on an initial context given by the previous state and action. Suppose the length of a state and action symbol together is $d$ and let $[x]$ denote the binary representation of a symbol $x$ and $[x]_i, [x]_{<i}$ denote the $i$-th bit and the first $i-1$ bits of $x$ respectively. Given the sequence $sra_{1:n}$, the distribution of the first bit is computed as a Bayesian mixture model by a depth $d$ context tree:

$$\hat{P}^{(1)}([sr_n]_1 | asr_{1:n-1}, a_n) = \sum_{T \in \mathcal{T}_d} 2^{-\gamma(T)} \hat{P}([sr_n]_1 | T([s_{n-1}a_n]), sr_{1:n-1})$$

Here $T([s_{n-1}a_n])$ denotes the leaf node in the PST $T$ that is reached by following the suffix of $[s_{n-1}a_n]$. Limiting the context to the previous state-action symbols encodes the Markov assumption into the modelling of the state-reward transition. The distribution $\hat{P}([sr_n]_1 | T([s_{n-1}a_n]), sr_{1:n-1})$ of $[sr_n]_1$ under the model $T$ is provided by a KT estimator [29] at $T([s_{n-1}a_n])$ that is computed by accumulating the frequency of bits following $[s_{n-1}a_n]$ from the history $sr_{1:n}$. The function $\gamma$ measures the complexity of a model via a prefix coding and the weighting term $2^{-\gamma(T)}$ provides an Occam-like penalty on the model $T$. For more details on the prior weighting see [4]. For the $l$-th bit, the distribution conditions on the previous state-action as well as the previous $l-1$ bits of $sr_n$:

$$\hat{P}^{(l)}([sr_n]_l | [sr_n]_{<l}, asr_{1:n-1}, a_n) = \sum_{T \in \mathcal{T}_{d+l-1}} 2^{-\gamma(T)} \hat{P}([sr_n]_l | T([s_{n-1}a_n][sr_n]_{<l}), sr_{1:n-1}) \quad (5)$$

The prediction for $sr_n$ using $\Phi$-BCTW is then given by composing the predictions for each bit:

$$\hat{P}(sr_n|asr_{1:n-1}, a_n) = \prod_{j=1}^{k} \hat{P}^{(j)}([sr_n]_j|[sr_n]_{<j}, sra_{1:n-1}, a_n)$$

Note that the conditional reward distribution $\hat{P}(r_n|sra_{1:n-1}, s_n)$ is modelled by the last $|[r_n]|$ context trees. The following result states that $\Phi$-BCTW computes a mixture environment model.

**Proposition 1.** Let $\hat{P}(sr_{1:n}|a_{1:n}, T) = \prod_{i=1}^{n} \prod_{j=1}^{k} \hat{P}([sr_i]_j|T([s_{i-1}a_i][sr_i]_{<j}), sr_{1:i-1})$ and $\Gamma(T) = \sum_{j=1}^{k} \gamma(T_i)$. $\Phi$-BCTW computes a mixture environment model of the form:

$$\xi(sr_{1:n}|a_{1:n}) = \sum_{T \in \mathcal{T}_d \times \ldots \times \mathcal{T}_{d+k-1}} 2^{-\Gamma(T)} \hat{P}(sr_{1:n}|a_{1:n}, T)$$

The following guarantees fast reward distribution convergence for mixture environment models.

**Theorem 3.** *Let $\mu$ be the true environment and $\xi$ be the mixture environment model over a model class $\mathcal{M}$. For all $n \in \mathbb{N}, a_{1:n}, o_{1:n}$*

$$\sum_{k=1}^{n} \sum_{r_{1:k}} \mu(r_{<k}|ao_{<k}) \left[\mu(r_k|aor_{<k}ao_k) - \xi(r_k|aor_{<k}ao_k)\right]^2 \leq \min_{\rho \in \mathcal{M}} \left[-\ln w_0^\rho + D_{1:n}(\mu||\rho)\right]$$

*where $D_{1:n}(\mu||\rho) \coloneqq \sum_{r_{1:n}} \mu(r_{1:n}|ao_{1:n}) \ln \frac{\mu(r_{1:n}|ao_{1:n})}{\rho(r_{1:n}|ao_{1:n})}$.*

The proofs of Proposition 1 and Theorem 3 can be found in Appendix A.

As a mixture environment model, $\Phi$-BCTW's conditional distribution over reward sequences is guaranteed to converge quickly by Theorem 3 provided the true environment can be modelled by the provided predicates. Combined with $\Phi$-BCTW's Occam-like model size penalty, $\Phi$-BCTW can be viewed as optimising $Cost_M^0$ over the set of abstractions defined by the given predicates.

### 3.3 Expectimax Approximation With $\rho$UCT

We use Monte-Carlo Tree Search to perform an approximation of the finite horizon expectimax operation in AIXI. We employ the $\rho$UCT algorithm [4] where actions are selected according to the UCB policy criteria. Unlike $\rho$UCT however, we are able to store search nodes of the tree and maintain value estimates for the duration of the agent's life-span. This is due to the feature selection and modelling by $\Phi$-BCTW reducing the environment model to an MDP with a 'small' state space. Thus, the number of samples through the search tree that need to be generated per step is greatly reduced.

## 4 Experiments

We evaluate our $\Phi$-AIXI-CTW agent's performance across a number of different domains. We begin by evaluating our agent's performance on four simpler domains before considering the epidemic control problem. All experiments were performed on a 12-Core AMD Ryzen Threadripper 1920x processor and 32 gigabytes of memory.

### 4.1 Simple Domains

We consider four simple domains: Biased rock paper scissors (RPS) [30], Taxi [31], Jackpot and Stop Heist. A full description of each of the environments is provided in Appendix C.1. As baseline comparison methods we compare against two decision-tree based, iterative state abstraction methods using splitting criteria as defined in U-Tree [32] and PARSS-DT [33]. In U-Tree, an existing node (representing a state) is split if splitting results in a statistically significant difference in the resulting Q-values as computed by a Kolmogorov-Smirnov test. In PARSS-DT, nodes are split if the resulting value functions are sufficiently far apart. Model learning is simply done by frequency estimation. To ensure our method does not have an informational advantage, both baseline methods are given the same initial set of predicate functions to consider node splits from. In both baselines, leaf nodes were tested for splits every 100 steps.

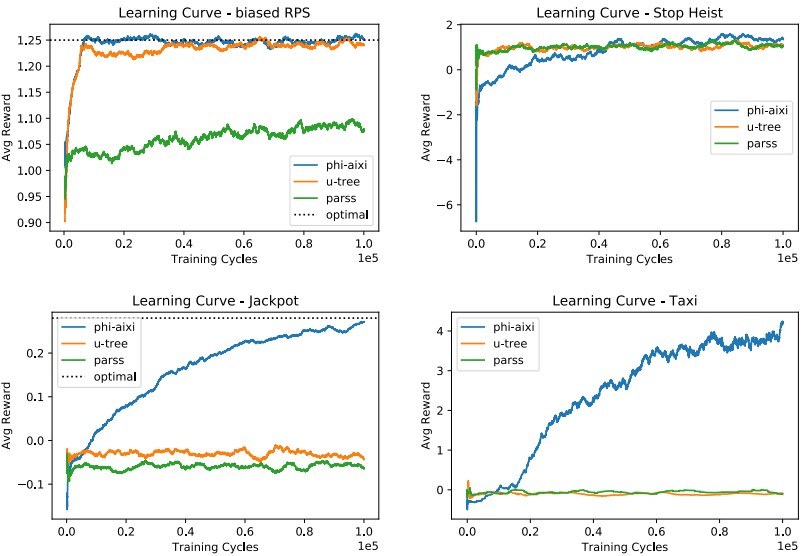

Figure 1: Learning curves on four domains.

From Figure 1, it can be seen that our $\Phi$-AIXI-CTW agent either outperforms or performs on par with the two baseline decision tree methods. U-Tree was able to perform well in the biased RPS and Stop Heist domains. The PARSS based method was able to perform well on the Stop Heist domain and was displaying evidence of learning, albeit slowly, on the biased RPS domain. U-Tree and PARSS were both unable to perform well in the Jackpot and Taxi environments. One possible issue is that Jackpot and Taxi require more predicates than biased RPS and Stop Heist to be able to perform well. Also, both methods can be susceptible to producing spurious splits. This can make the state space too large and the resulting transition model can become too difficult to learn with.

## 4.2 Epidemic Control over Contact Networks

We now evaluate the agent's performance on the novel task of learning to control an epidemic process over large-scale contact networks, a highly structured environment that exhibits an exponential state space, partial observability and history dependency.

**Epidemic Processes on Contact Networks.** In this paper epidemic processes are modelled on contact networks [34, 35, 36]. A contact network is an undirected graph where the nodes represent individuals and the edges represent interactions between individuals. Each individual node in the graph is labelled by one of four states corresponding to their infection status: Susceptible (S), Exposed (E), Infectious (I), and Recovered (R). Every node also maintains an immunity level $\omega$. The model evolves as a POMDP. At time $t$, the temporal graph is given by $G_t = (V, \mathcal{E}_t)$ with $V$ as the set of nodes and $\mathcal{E}_t$ as the set of edges at time $t$. A function $\zeta_t : V \to \{S, E, I, R\} \times \{1, \eta_1, \eta_2\}$ maps each node to its label and one of three immunity levels where $\eta_1, \eta_2 \in \mathbb{R}_+$. Together, $(G_t, \zeta_t)$ constitute the state. At time $t$, a Susceptible node with $k_t$ Infectious neighbours becomes Exposed with probability $\frac{1-(1-\beta)^{k_t}}{\omega}$, where $\beta \in [0, 1]$ is the rate of transmission. An Exposed node becomes Infectious at rate $\sigma$. Similarly, an Infectious node becomes Recovered at a rate $\gamma$ and becomes Susceptible again at a rate $\rho$.

At every time step, the process emits an observation on each node from $\{+, -, ?\}$ corresponding to whether a node tests positive, negative or is unknown/untested. The observation model is parametrised such that positive tests are more likely if the underlying node is Infectious, corresponding to the realistic scenario where sick individuals are more likely to test and also test positive. A more detailed exposition on the exact transition and observation models is given in Appendix B.

The agent can perform an action at each time step. We consider a set of 11 possible actions $\{DoNothing, Vaccinate(i,j), Quarantine(i)\}$, where $i \in [0, 0.2, 0.4, 0.6, 0.8, 1.0]$ and $j = i + 0.2$. A $Vaccinate(i,j)$ action increases the immunity level of the top $i$th to $j$th percent of nodes as ranked by betweenness centrality by one level, up to the maximum value. Note that a conferred immunity

level lasts for the entire length of an episode unless increased. A $Quarantine(i)$ action quarantines the top $i$th percent of nodes by removing all edges incident on those nodes for one time step. The reward function is parametrised to evaluate the agent's ability to balance the cost of its actions against the cost of the epidemic. At each time step, the instantaneous reward is given by

$$r_t(o_t, a_{t-1}) := -\lambda Positives(o_t) - Action\_Cost(a_{t-1})$$

where $\lambda \in R_+$, $Positives(o_t)$ counts the number of positive tests in the observation $o_t$ and $Action\_Cost(a_{t-1})$ is a function determining the cost of each action. Thus changing $\lambda$ changes the ratio of contributions to the reward received at each time step. If the agent successfully terminates the epidemic, i.e. there are no more Exposed or Infectious nodes, the agent receives a positive reward of 2 per node. During a run, it is possible the environment never reaches an absorbing state. If the agent has not successfully stopped the epidemic process after 1000 steps, we terminate the episode.

Epidemic control is a topical subject given the prevalance of COVID-19 and recent approaches to this question vary wildly in terms of the how the problem is modelled. Most approaches vary in the different ways they model the dynamics of an epidemic and the action space considered is usually fairly small [37, 38, 39, 40, 41, 42, 43]. Ultimately, the differences in the modelling of epidemic processes makes comparison between results difficult. With no consensus on the appropriate model to use, our model is chosen as it is sufficiently complex to demonstrate the efficacy of our agent.

**Experimental Setup.** We use an email network dataset as the underlying contact network, licensed under a Creative Commons Attribution-ShareAlike License, containing 1133 nodes and 5451 edges [44, 45]. The transition model, observation model, $Action\_Cost(a_t)$ are parametrised the same way across all experiments (see Table 1 in Appendix B). A $Quarantine(i)$ action imparts a cost of 1 per node that is quarantined at the given time step. A $Vaccinate(i, j)$ action imparts a lower cost of 0.5 per node. The parameters $\lambda, \eta_1, \eta_2$ are varied across experiments. We generate a set of 1489 predicate functions consisting of predicates on top of functions such as the percentage each action was selected in the last $N$ steps, the observed infection rate on different subsets of nodes, and also the infection rate on different subset of nodes as computed by particle filtering. A full description of how the predicates were generated is given in Appendix D. The $\Phi$-AIXI-CTW agent is trained in an online fashion. The agent explores with probability $\epsilon\alpha^t$ at each step $t$ until $\epsilon\alpha^t < 0.03$, where the agent performs in an $\epsilon$-greedy way with exploration rate 0.03. RF-BDD was performed with a threshold value of 0.9 across all rewards.

### 4.2.1  Results

**Selected Features.** The set of predicates chosen in each experiment by RF-BDD are described in Appendix D. In general, RF-BDD tended to select predicates representing bits in the binary representation of the observed infection rate and the rate of change of the observed infection rate on different subsets of the betweenness centrality nodes. A random selection of different action sequence indicator functions were also kept. Note that predicates on the infection rate as computed by particle filtering were not selected and this can be attributed to the particle degeneracy issue that occurs in high dimensional problems [46]. These choices by RF-BDD highlight its ability to select a small set of predicates that provided useful information for epidemic control. The inclusion of a random selection of different action sequence indicator functions also demonstrates that it is not perfect. A more thorough investigation is the topic of future work.

**Comparison to Baseline Methods.** Both U-Tree and PARSS were used as baseline methods in the same configuration as in the simpler domains. From Figure 2, $\Phi$-AIXI-CTW outperforms the two baseline methods and both U-Tree and PARSS were unable to improve. The epidemic problem as formulated is likely difficult for both U-Tree and PARSS as the number of predicates required to perform well is fairly large. Also, both methods can be susceptible to producing spurious splits. This can make the resulting state space and transition model too large and difficult to learn with.

**RF-BDD vs. Random Forest Feature Selection.** We also compare the performance of our agent under RF-BDD or Random Forest feature selection. Other feature selection methods such as forward selection methods [47] were also considered but were found to offer no advantage over Random Forest. For each experiment, we plot four graphs. The first graph plots the learning curve computed as a moving average over 5000 steps of the reward per cycle. The remaining graphs provide insight into the learnt policy. The second graph plots the percentage each type of action is performed over a moving average of 500 steps as well as a moving average of the infection rate over the same window.

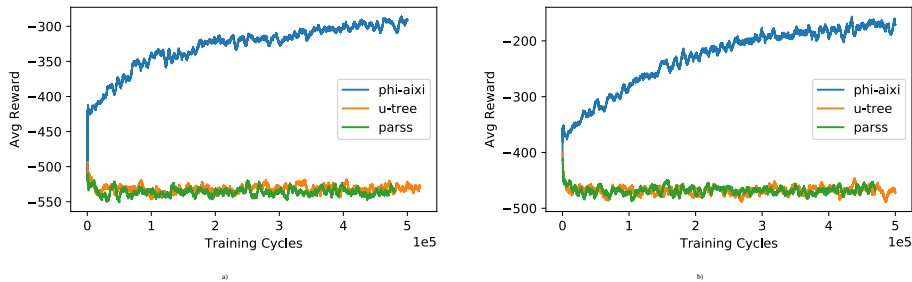

Figure 2: Baseline comparison results on a) $\lambda = 1, \eta_1 = 2, \eta_2 = 4$. b) $\lambda = 1, \eta_1 = 10, \eta_2 = 20$.

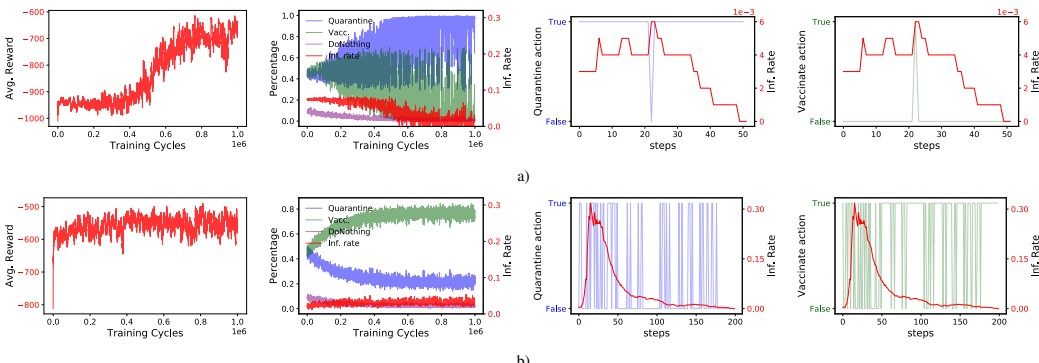

Figure 3: $\Phi$-AIXI-CTW agent with RF-BDD feature selection. a) Experiment 1: $\lambda = 10, \eta_1 = 2, \eta_2 = 4$. b) Experiment 2: $\lambda = 10, \eta_1 = 10, \eta_2 = 20$.

The remaining two graphs provide an example of the agent's behaviour and its impact on the infection rate over an episode collected toward the end of training.

Figure 3 depicts the $\Phi$-AIXI-CTW agent's performance with RF-BDD feature selection over two experiments with differing vaccination efficacy and an infection rate weighting of $\lambda = 10$. RF-BDD was run with 500 trees generated using random samples of 8 features and used a threshold value of 0.9. In both experiments, the agent learns to be more aggressive in terminating the epidemic. In Experiment 1 the agent's behaviour is to quarantine almost always, and occasionally vaccinate. This is reflected in the plots of the agent's behaviour in an episode toward the end of training. Quarantine actions are performed the entire time except once, resulting in a fast termination of the epidemic process. In contrast, Experiment 2 shows that the agent prefers to vaccinate more often. This preference can be attributed to the added effectiveness that a stronger vaccination action imparts to lowering the infection rate. When analysing the agent's behaviour over an episode, it is clear that the agent chooses to vaccinate almost always whilst only quarantining at timely moments. This results in the agent also learning to terminate the episode early, albeit not as quickly. In comparison to Experiment 1, the average reward per cycle is also higher at the end of learning.

Figure 4 depicts the $\Phi$-AIXI-CTW agent's performance with Random Forest feature selection. Random Forest was initialized with 500 trees and splits were chosen using either the gini impurity or entropy and the top thirty highest weighting features were chosen. The resulting predicate sets were very similar for the two criteria and so we only present results for the entropy criteria case. When comparing the agent's performance when using RF-BDD (Figure 3) versus Random Forest (Figure 4), it is quite clear that the agent performs better with the predicates chosen using RF-BDD. As can be seen from Figure 4a) and 4b), the agent's learning curve when using the predicates chosen by Random Forest did not improve in either case and actually decreased. The remaining three plots also show that there is no clear discernible pattern of behaviour learnt by the agent. This suggests that the features chosen by Random Forest were not informative enough to allow the agent to learn to perform well. The inability of traditional feature selection methods like Random Forest to pick out useful features for the agent was a strong motivation for constructing an alternative in RF-BDD.

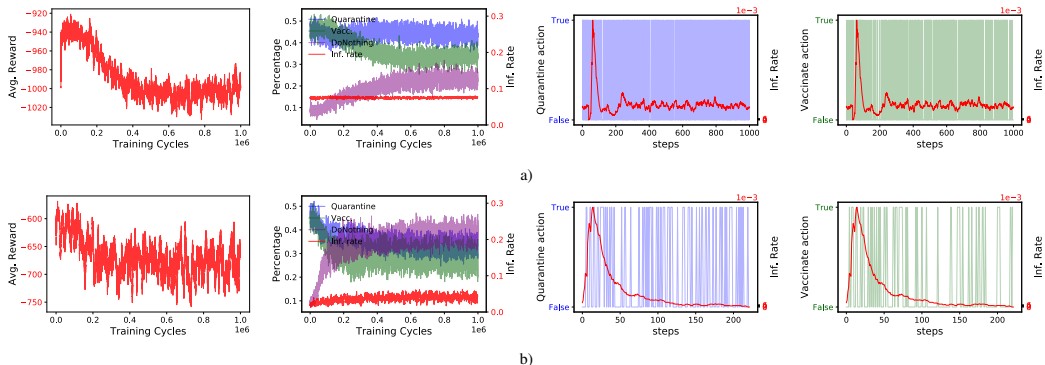

Figure 4: Φ-AIXI-CTW agent with Random Forest feature selection. a) Experiment 3: $\lambda = 10, \eta_1 = 2, \eta_2 = 4$. b) Experiment 4: $\lambda = 10, \eta_1 = 10, \eta_2 = 20$.

Overall, the results demonstrate that the Φ-AIXI-CTW agent is able to learn plausible behaviours under the different environment settings. This demonstrates the potential of our approach in reducing a complex environment to a simpler problem whilst still maintaining performance. Nevertheless, it is also quite clear that the agent did not learn optimal behaviour except potentially in the case of Experiment 1. In Experiment 2, the agent did not learn that vaccinating a node more than twice does not further reduce the cost received. However, this is not surprising as this is a difficult causal relationship to learn from data alone. Furthermore, the full set of generated predicates do not contain enough information to isolate this particular history dependency. In fact, the predicates selected generally capture different statistics about the infection rate at the current time step. If the value estimate of vaccinating in a highly visited state is high due to it lowering future received costs, the agent will learn to perform the vaccination action every time the state is encountered, even if it has done so many times before. We expect that importing better domain knowledge into the original set of predicates would solve this issue for the Φ-AIXI-CTW agent.

## 5 Conclusion

We have introduced the Φ-AIXI-CTW agent model, an agent that expands the model class that AIXI may be approximated over to structured and non-Markovian domains via logical state abstractions. We show that our approach is competitive on simple domains and is sufficiently powerful to handle control of epidemic processes on contact networks, a class of problems with a highly structured exponential state space, partial observability and history dependency. In doing so, our result provides another piece of strong evidence that AIXI is not just a mathematically interesting theory, but can eventually lead to practically useful agents.

Whilst RF-BDD has demonstrated its usefulness empirically in our experiments, a better understanding of how it works theoretically and how it behaves in other contexts is important. In [48] the authors identify the epidemic control problem within a class of problems they call the control of diffusion processes over networks through nodal interventions. The epidemic control problem constitutes an instance of this class of problems and it would be interesting to extend our agent setup to other problems within this domain.

A key concern for the real-world application of reinforcement learning on contact networks for control is the preservation of privacy for individual nodes. Our agent setup does not restrict the usage of any form of domain knowledge on the complete graph such as an individual node's contact network and extended neighbourhood. Investigating the performance of our agent under privacy preserving constraints such as differential privacy [49] is an interesting topic for future research.

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
