# A  Φ-BCTW Results

The following proves Proposition 1.

*Proof.* Under Φ-BCTW, the conditional probability of $sr_{1:n}$ given the actions $a_{1:n}$ can be expressed as

$$\xi(sr_{1:n}|a_{1:n}) = \prod_{i=1}^{n} \hat{P}(sr_i|asr_{1:i-1}, a_i)$$

$$= \prod_{i=1}^{n}\prod_{j=1}^{k} \hat{P}^{(j)}([sr_i]_j|[sr_i]_{<j}, asr_{1:i-1}, a_i)$$

$$\stackrel{(a)}{=} \prod_{i=1}^{n}\prod_{j=1}^{k}\left(\sum_{T\in\mathcal{T}_{d+j-1}} 2^{-\gamma(T)}\hat{P}([sr_i]_j|T([s_{i-1}a_i][sr_i]_{<j}), sr_{1:i-1})\right)$$

$$= \sum_{T_1\in\mathcal{T}_d}\cdots\sum_{T_k\in\mathcal{T}_{d+k-1}}\left(\prod_{j=1}^{k} 2^{-\gamma(t)}\right)\left(\prod_{j=1}^{k}\prod_{i=1}^{n}\hat{P}([sr_i]_j|T([s_{i-1}a_i][sr_i]_{<j}), sr_{1:i-1})\right)$$

$$\stackrel{(b)}{=} \sum_{T\in\mathcal{T}_d\times\ldots\times\mathcal{T}_{d+k-1}} 2^{-\Gamma(T)}\hat{P}(sr_{1:n}|a_{1:n}, T).$$

Note that (a) follows from Equation 5 and (b) follows by definition. □

The following proof of Theorem 3 follows the same steps as Theorem 1 in [4].

*Proof.*

$$\sum_{k=1}^{n}\sum_{r_{1:k}} \mu(r_{<k}|ao_{<k})\left(\mu(r_k|aor_{<k}ao_k) - \xi(r_k|aor_{<k}ao_k)\right)^2$$

$$= \sum_{k=1}^{n}\sum_{r_{<k}} \mu(r_{<k}|ao_{<k})\sum_{r_k}\left(\mu(r_k|aor_{<k}ao_k) - \xi(r_k|aor_{<k}ao_k)\right)^2$$

$$\stackrel{(a)}{\leq} \sum_{k=1}^{n}\sum_{r_{<k}} \mu(r_{<k}|ao_{<k})\sum_{r_k}\mu(r_k|aor_{<k}ao_k)\ln\frac{\mu(r_k|aor_{<k}ao_k)}{\rho(r_k|aor_{<k}ao_k)}$$

$$= \sum_{k=1}^{n}\sum_{r_{1:k}} \mu(r_{1:k}|ao_{1:k})\ln\frac{\mu(r_k|aor_{<k}ao_k)}{\xi(r_k|aor_{<k}ao_k)}$$

$$= \sum_{k=1}^{n}\sum_{r_{1:n}} \mu(r_{1:n}|ao_{1:n})\ln\frac{\mu(r_k|aor_{<k}ao_k)}{\xi(r_k|aor_{<k}ao_k)}$$

$$= \sum_{r_{1:n}} \mu(r_{1:n}|ao_{1:n})\sum_{k=1}^{n}\ln\frac{\mu(r_k|aor_{<k}ao_k)}{\xi(r_k|aor_{<k}ao_k)}$$

$$\stackrel{(b)}{=} \sum_{r_{1:n}} \mu(r_{1:n}|ao_{1:n})\ln\frac{\mu(r_{1:n}|ao_{1:n})}{\xi(r_{1:n}|ao_{1:n})}$$

$$= \sum_{r_{1:n}} \mu(r_{1:n}|ao_{1:n})\ln\frac{\mu(r_{1:n}|ao_{1:n})}{\rho(r_{1:n}|ao_{1:n})} + \sum_{r_{1:n}} \mu(r_{1:n}|ao_{1:n})\ln\frac{\rho(r_{1:n}|ao_{1:n})}{\xi(r_{1:n}|ao_{1:n})}$$

$$\stackrel{(c)}{\leq} D_{1:n}(\mu||\rho) + \sum_{r_{1:n}} \mu(r_{1:n}|ao_{1:n})\ln\frac{\rho(r_{1:n}|ao_{1:n})}{w_0^{\rho}\rho(r_{1:n}|ao_{1:n})}$$

$$= D_{1:n}(\mu||\rho) - \ln w_0^{\rho}$$

The inequality in (a) follows as $\sum_i (y_i - z_i)^2 \leq \sum_i y_i \ln \frac{y_i}{z_i}$ for $y_i, z_i \geq 0$, $\sum_i y_i = 1$, $\sum_i z_i = 1$. The equality in (b) follows by definition as $\xi(r_{1:n}|ao_{1:n}) := \prod_{k=1}^n \xi(r_k|aor_{<k}ao_k)$. Finally, (c) follows by definition of a mixture environment model. Note that a $\Phi$-BCTW mixture environment model is a model where $\xi(r_k|aor_{<k}ao_k) = \hat{p}(r_k|sra_{<k}sa_k)$. $\qquad\square$

## B   Dynamics for epidemic processes on contact networks

$$S \underset{\kappa}{\overset{\rho}{\rightleftarrows}} E \xrightarrow{\sigma} I \xrightarrow{\gamma} R$$

Figure 5: SEIRS compartmental model on contact networks with transmission rate $\kappa = \frac{1-(1-\beta)^{k_t}}{\omega}$ (where $\beta$ is the contact rate), latency rate $\sigma$, recovery rate $\gamma$ and loss of immunity rate $\rho$.

Recall that the state of the epidemic process on contact networks is given by $(G_t, \zeta_t)$ at time $t$ where $G_t = (V, \mathcal{E}_t)$ is the temporal graph at time $t$ and $\zeta_t : V \to \{S, E, I, R\} \times \{1, \eta_1, \eta_2\}$ maps each node to its label and one of three immunity levels. Let $\zeta_t(v) = (\zeta_{1,t}(v), \zeta_{2,t}(v))$ where $\zeta_{1,t}$ maps a node to its infection status and $\zeta_{2,t}$ maps a node to its immunity level. The agent performs an action $a_t$. Quarantine actions modify the underlying connectivity of the graph $G_t$ by removing any edges that contain a quarantined node; this leads to the graph at time $t+1$ $G_{t+1}$. Vaccinate actions modify $\zeta_{2,t}$ such that vaccinated nodes have updated immunity levels in $\zeta_{2,t+1}$. The infection status label of every node evolves according to the following equations:

$$\tau(\zeta_{1,t+1}(v)|\zeta_{1,t}(v), s_t) = \begin{cases} \frac{1-(1-\beta)^{k_t}}{\zeta_{2,t}(v)} & \text{if } \zeta_{1,t+1}(v) = E \text{ and } \zeta_{1,t}(v) = S \\ 1 - \frac{1-(1-\beta)^{k_t}}{\zeta_{2,t}(v)} & \text{if } \zeta_{1,t+1}(v) = S \text{ and } \zeta_{1,t}(v) = S \\ \sigma & \text{if } \zeta_{1,t+1}(v) = I \text{ and } \zeta_{1,t}(v) = E \\ 1 - \sigma & \text{if } \zeta_{1,t+1}(v) = E \text{ and } \zeta_{1,t}(v) = E \\ \gamma & \text{if } \zeta_{1,t+1}(v) = R \text{ and } \zeta_{1,t}(v) = I \\ 1 - \gamma & \text{if } \zeta_{1,t+1}(v) = I \text{ and } \zeta_{1,t}(v) = I \\ \rho & \text{if } \zeta_{1,t+1}(v) = S \text{ and } \zeta_{1,t}(v) = R \\ 1 - \rho & \text{otherwise,} \end{cases} \qquad (6)$$

where $k_t$ denotes the number of Infectious, connected neighbours that $v$ has at time $t$.

The observations resemble the testing for an infectious disease with positive $+$ and negative $-$ outcomes. Recall that the observations on each node are from $\mathcal{O} = \{+, -, ?\}$ where $?$ indicates the corresponding individual has unknown/untested status. At time $t$, node $v \in V$ emits an observation according to the following distribution:

$$
\begin{aligned}
\xi_t^v(+|\zeta_{1,t}(v) = S) &= \alpha_S \mu_S, & \xi_t^v(+|\zeta_{1,t}(v) = E) &= \alpha_E \mu_E, \\
\xi_t^v(-|\zeta_{1,t}(v) = S) &= \alpha_S (1 - \mu_S), & \xi_t^v(-|\zeta_{1,t}(v) = E) &= \alpha_E (1 - \mu_E), \\
\xi_t^v(?|\zeta_{1,t}(v) = S) &= 1 - \alpha_S, & \xi_t^v(?|\zeta_{1,t}(v) = E) &= 1 - \alpha_E, \\
\\
\xi_t^v(+|\zeta_{1,t}(v) = I) &= \alpha_I \mu_I, & \xi_t^v(+|\zeta_{1,t}(v) = R) &= \alpha_R \mu_R, \\
\xi_t^v(-|\zeta_{1,t}(v) = I) &= \alpha_I (1 - \mu_I), & \xi_t^v(-|\zeta_{1,t}(v) = R) &= \alpha_R (1 - \mu_R), \\
\xi_t^v(?|\zeta_{1,t}(v) = I) &= 1 - \alpha_I, & \xi_t^v(?|\zeta_{1,t}(v) = R) &= 1 - \alpha_R.
\end{aligned}
\qquad (7)
$$

Here, $\alpha_S, \alpha_E, \alpha_I, \alpha_R$ denote the fraction of Susceptible, Exposed, Infectious and Recovered individuals, respectively, that are tested on average at each time step. The parameters $\mu_S, \mu_E, \mu_I, \mu_R$ denote the probability that a node that is Susceptible, Exposed, Infectious, Recovered respectively tests positive. Table 1 lists the transition and observation model parameters that were used in all experiments.

| $\beta$ | $\sigma$ | $\gamma$ | $\rho$ | $\alpha_S$ | $\alpha_E$ | $\alpha_I$ | $\alpha_R$ | $\mu_S$ | $\mu_E$ | $\mu_I$ | $\mu_R$ |
|---|---|---|---|---|---|---|---|---|---|---|---|
| 0.2 | 0.3 | 0.08 | 0.1 | 0.1 | 0.1 | 0.8 | 0.05 | 0.1 | 0.9 | 0.9 | 0.1 |

Table 1: Transition and observation model parameters.

# C    Extra Experimental Results

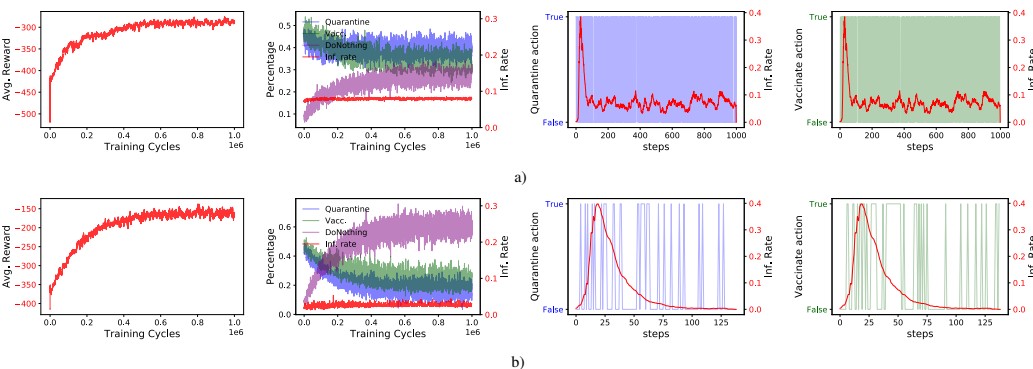

Figure 6: a) Experiment 5 $\lambda = 1, \eta_1 = 2, \eta_2 = 4$. b) Experiment 6 $\lambda = 1, \eta_1 = 10, \eta_2 = 20$.

Figure 6 depicts the agents behaviour over two experiments with differing vaccination efficacy and an infection rate weighting of $\lambda = 1$. With $\lambda = 1$, an observed positive node is equivalent in cost to quarantining a node or vaccinating two nodes. In Experiment 5 (Figure 6a)), the agent learns to perform quarantine and vaccination actions almost equally frequently and slightly more often that doing nothing. Whilst there is no clear discernible pattern in the agent's policy and it appears almost random, the learning curve shows that the agent has learnt to perform better than random. In Experiment 6 the agent prefers to do nothing most of the time and quarantine/vaccinate less. This difference between Experiments 5 and 6 can be attributed to the increased efficacy of vaccination actions in reducing the infection rate throughout an episode. In comparison to Experiments 1 and 2 (Figure 3), where $\lambda = 10$, the action percentages indicate that the agent is more conservative; the agent tends to be less active in managing the epidemic and is more likely to do nothing. This difference in behaviour directly reflects the different cost function between the two sets of experiments.

## C.1    Additional Domains

**Biased rock paper scissors (RPS).** This domain is taken from [30]. Biased RPS is an environment where the agent repeatedly plays RPS against the environment which has a slight bias in its strategy. The environment plays randomly unless it won the previous round playing rock. In this case, the environment will always play rock at the current time step. This bias can be simply captured by a predicate $IsRockAndLose_{t-1}(h)$ which returns whether the environment played rock and the agent lost at the previous time step. A set of 1000 predicates were generated including predicates that detect whether the nth bit of the suffix of the history equals 1, $Suffix_N \circ (= 1)(h)$, and uninformative predicates such as returning whether a random bit equals 1.

**Stop Heist.** The Stop Heist environment is an environment exhibiting history dependency where the agent acts as the surveillance team for a bank. At every time step, the agent receives a binary observation representing whether a known suspect arrives at the bank. The suspect's arrival times are determined by a Hawkes process. As a Hawkes process is self-exciting, recent arrivals increase the chance of arriving in the near term. The chance that a suspect performs a heist is also modelled as a self-exciting process, increasing the more frequently the suspect arrives. The agent has two actions it can perform: Do Nothing or Stop Suspect. The agent incurs a penalty of -100 if the suspect performs a heist and the agent chose to Do Nothing. If the agent chooses to Stop Suspect and the suspect was performing a heist, it receives a reward of 100. If the agent performs Stop Suspect but the suspect did not plan to perform a heist, it receives a minor penalty of -1. Otherwise, the agent receives 0 reward. This environment requires the agent to capture predicates that maintain some knowledge of the history of arrivals. To facilitate this, predicates of the following form were provided:

- $Percent_{o,n} \circ (\geq p)(h)$ for various values of $o \in \mathcal{O}, n \in \mathbb{N}$ and $p \in [0, 1]$.

$Percent_{o,n}$ computes the percentage of times that observation $o$ was received in the past $n$ steps. $(\geq p)$ returns whether the provided argument is greater than $p$. A set of 1000 predicates were generated

for various values of $n$ and $p$ as well as including uninformative predicates such as returning whether a random bit equals 1.

**Jackpot.** The Jackpot environment is an environment where the agent plays a betting game against the environment at every time step. At every time step, the agent can choose to either Bet or Pass. If the agent performs Pass, it receives 0 reward. The environment is initiated with a pre-defined list of numbers. If the agent chooses to Bet on a time step that is a multiple of a number in the pre-defined list of numbers, it receives reward 1 with 70% chance. If instead the agent chooses to Bet on a time step that is not a multiple of a number in the pre-defined list of numbers, then the agent receives reward -1 with 70% chance. The observation received at every time step is constant and is uninformative. This environment requires the agent to capture predicates that provide some knowledge about arithmetic and also be able to count the time steps. Predicates of the following form were provided:

- $Count \circ IsMultiple_j(h)$ for various values of $j \in \mathbb{N}$.

$Count$ returns the number of steps that have occurred in the provided history $h$ and $IsMultiple_j$ returns whether the provided argument is a multiple of $j$. A set of 1000 predicates were generated for various values of $j \in \mathbb{N}$ as well as including uninformative predicates such as returning whether a random bit equals 1.

**Taxi.** The Taxi environment is the well-known environment first introduced in [31]. The agent acts as a Taxi in a grid world and must move to pick up a passenger and drop the passenger off at their desired destination. Instead of the 5x5 grid traditionally considered, we consider a 2x5 grid with no intermediate walls. We also change the reward for dropping the passenger at its destination successfully to 100. These modifications were made to shrink the planning horizon as well as make the original problem less sensitive to parameters for the three algorithms tested. Four destination locations are still used. The predicates available to the agent are suffix predicates that determine whether different bits of the history sequence are equal to 1. The agent will be able to recover the original MDP if it captures the suffix predicates that comprise the last observation received.

# D   Predicates

We assume that the agent has a background theory $\Gamma$ consisting of:

- a graph $G = (V, \mathcal{E})$ that captures, either exactly or approximately, the structure of the initial graph $G_0$ but not the disease status of nodes (the connectivity between people could be inferred from census data, telecommunication records, contact tracing apps, etc);
- the transition and observation functions of a dynamics model (i.e. Equations 6 and 7 for an SEIRS model) of the underlying disease but not the parameters. This information would be provided by experts in the chosen domain, i.e. epidemiologists working on epidemic modelling.

The agent has access to the following functions:

- $Encode_i$ splits the possible range of its argument into $2^i$ equal sized buckets and encodes its argument by the number of whichever bucket it falls into. If the range is unbounded, it is first truncated.
- $Bit_i$ takes a bit string and returns the $ith$ bit.
- $NaiveInfectionRate_{t,\nu}$ takes a history sequence $h \in \mathcal{H}$ and computes the infection rate at time $t$ over the set of nodes $\nu \subseteq V$ as the observed infection rate plus a constant multiplied by the number of nodes observed as unknown.
- $InfectionRateOfChange_{t,\nu}$ takes a history sequence $h \in \mathcal{H}$ and computes the change in infection rate between timesteps $t-1$ and $t$ over the set of nodes $\nu \subseteq V$.
- $PercentAction_{a,N}$ takes a history and returns the percentage of time action $a$ was selected in the last $N$ timesteps.
- $ActionSequenceIndicator_{a_{1:k}}$ is an indicator function returning 1 if the last $k$ actions performed match $a_{1:k}$ and 0 otherwise.

- $MAReward_w$ takes a history and returns the moving average of the reward over a window of size $w$.
- $RateOfChange$ takes two real numbers and computes the ratio between them.
- $ParticleFilter_{\theta,M}$ takes a history sequence and approximates the belief state using the transition and observation models given by $\theta$ and $M$ particles.
- $ParticleInfRate$ takes a belief state represented by a set of particles and computes the expected infection rate.

Function composition is handled by the (reverse) composition function

$$\circ : (a \to b) \to (b \to c) \to (a \to c)$$

defined by $((f \circ g)\ x) = (g\ (f\ x))$.

The set of predicates generated using the above functions were of the following form:

(1) $NaiveInfectionRate_{t,\nu} \circ Encode_5 \circ Bit_i \circ (=1)(h)$  for various $i, \nu \subseteq V,\ t = 1$

(2) $InfectionRateOfChange_{t,\nu} \circ Encode_7 \circ Bit_i \circ (=1)(h)$  for various $i, \nu \subseteq V,\ t = 1$

(3) $PercentAction_{a,N} \circ Encode_8 \circ Bit_i \circ (=1)(h)$  for all $a \in \mathcal{A}$ and various values of $N$

(4) $ActionSequenceIndicator_{a_{1:k}}(h)$  for various $a_{1:k},\ k \geq 1$

(5) $\lambda s.RateOfChange(MAReward_{w_1}(s), MAReward_{w_2}(s)) \circ (\geq 1)(h)$ for $w_1, w_2 \in \mathbb{N}$

(6) $ParticleFilter_{\theta,M} \circ ParticleInfRate \circ Encode_5 \circ Bit_i \circ (=1)(h)$ for various $\theta, M = 100$

The choice of subsets of nodes were limited to 20% percentile ranges of the nodes ranked by betweenness centrality and degree centrality. Note that retrieving a subset of the bits from an encoded value essentially selects a range that the encoded value falls within. For example, suppose we have two bits encoding values from 0 to 3. The largest bit indicates whether the encoded value is greater than 2 or not.

Table 2 lists the number of each type of predicate that were generated in the initial set. Table 3 details the type of predicates that were chosen in each experiment.

| Predicate type | Number generated |
|---|---|
| (1) | 55 |
| (2) | 80 |
| (3) | 693 |
| (4) | 633 |
| (5) | 8 |
| (6) | 20 |

Table 2: Number of predicates generated of each type. In total there are 1489 predicates generated.

| | | Predicate Type | | | | | | |
|---|---|---|---|---|---|---|---|---|
| **Experiment** | **FS** Method | **(1)** | **(2)** | **(3)** | **(4)** | **(5)** | **(6)** | **Total** |
| 1 | RF-BDD | 8 | **7** | 17 | 2 | 0 | 0 | 34 |
| 2 | RF-BDD | 3 | **9** | 9 | 2 | 0 | 0 | 23 |
| 3 | Random Forest | 16 | 0 | 6 | 0 | **8** | 0 | 30 |
| 4 | Random Forest | 10 | 0 | 12 | 0 | **8** | 0 | 30 |
| 5 | RF-BDD | 6 | **9** | 9 | 5 | 0 | 0 | 29 |
| 6 | RF-BDD | 10 | **4** | 16 | 4 | 0 | 0 | 34 |

Table 3: Number of predicates generated of each type for each experiment.

As can be seen from Table 3, the RF-BDD feature selection method was able to pick out predicates that provided information about the current infection rate (type (1)), its rate of change (type (2)), action percentages (type (3)) and a few action sequence indicators (type (4)). This behaviour was consistent across all experiments where RF-BDD was used.

Random Forest feature selection chose predicates that provided information about the current infection rate (type (1)), action percentages (type (3)), and the reward rate of change (type (5)).

Note that predicates on the infection rate as computed by particle filtering were not selected by either method. This can be attributed to the particle degeneracy issue that occurs in high dimensional problems [46]. When the size of the state space is very high dimensional and the number of particles used is not sufficient, the particle set can collapse to a very small number of points when updated, resulting in poor estimates.

It is likely that the difference in performance when comparing Experiments 1, 2, 3, and 4 (see Appendix C stems from the fact that RF-BDD chose type (2) predicates whereas Random Forest chose type (5) predicates. It is unlikely that type (4) predicates provided much useful information as the sequences chosen were not insightful and varied across the four experiments. In contrast, type (2) predicates inform the agent about the infection rate of change, which can provide useful information about the next state of the epidemic process provided the transition is sufficiently smooth.

# E  Feature Selection Using Binary Decision Diagrams

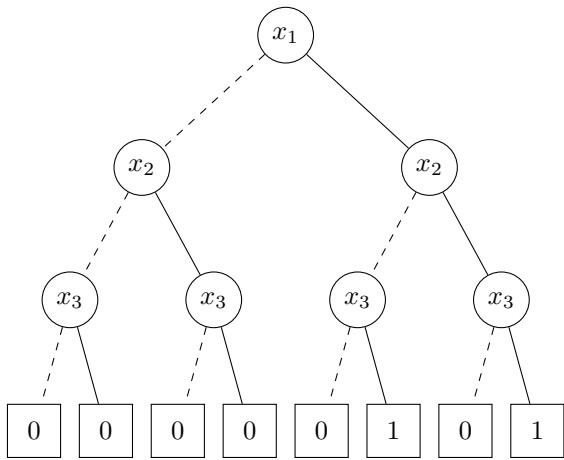

Figure 7: Full binary tree representation of the boolean function $f(x_1, x_2, x_3) = 1$ if $x_1$ and $x_3$.

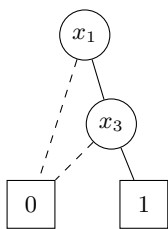

Figure 8: Reduced BDD for the boolean function $f(x_1, x_2, x_3) = 1$ if $x_1$ and $x_3$.

Figure 7 depicts the full binary tree representation of the boolean function $f(x_1, x_2, x_3) = 1$ if $x_1$ and $x_3$. Dotted lines indicate a path that is traversed when the boolean variable at the node equals 0 and solid lines are traversed when the boolean variable equals 1. We will always place variables $x_i$ higher in the tree than variables $x_j$ if $i < j$. Figure 8 depicts the same Boolean function but in reduced BDD form; as can be seen, the redundant feature $x_2$ has been eliminated.

We now consider the properties of reduced BDDs that make them appropriate for feature extraction. Every Boolean function $f : \{0, 1\}^n \rightarrow \{0, 1\}$ corresponds to a $2^n$ bit string representing the function's output on a canonical ordering of its input. More specifically, let $\beta$ be the $2^n$ bit string representing $f$. Then $\beta$ starts with $f(0, \ldots, 0)$ and continues with $f(0, \ldots, 0, 1)$, $f(0, \ldots, 1, 0)$, $f(0, \ldots, 1, 1)$, ..., until $f(1, \ldots, 1, 1)$. The $2^n$ bit string is known as a Boolean function's truth table. For example, the truth table of $f(x_1, x_2, x_3) = 1$ if $x_1$ and $x_3$ is 00000101. The truth table

also corresponds to the sequence of outputs we would get from the BDD representing the Boolean function if we traversed the BDD by taking 0 paths before 1 paths. We now define the notion of a bead.

**Definition 3** (Bead). A *bead* of order $n$ is a truth table $\beta$ such that there does not exist a string $\alpha$ of length $2^{n-1}$ such that $\beta = \alpha\alpha$.

In other words, a bead is a truth table that does not repeat itself at its mid-point. Now note that every node in a BDD can be identified with a substring of the truth table. Consider the full binary tree in Figure 7 again. The root node of the BDD can be identified with the full truth table of length $2^3$. The nodes one layer below correspond to a length $2^2$ partition of of the truth table. The full identification is shown in Figure 9. Depending on the truth table, nodes in a non-reduced BDD can correspond to either beads and non-beads. A key property of reduced BDD structures is that they only maintain nodes corresponding to beads.

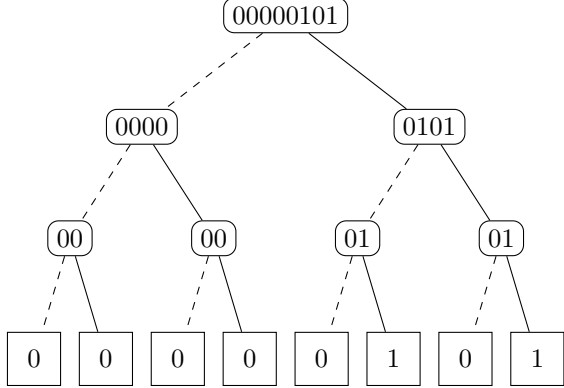

Figure 9: Full binary tree of the $f(x_1, x_2, x_3) = 1$ if $x_1$ and $x_3$ with nodes identified with corresponding beads.

**Theorem 4.** *Suppose $G = (V, E)$ is a reduced BDD representing a Boolean function $f$ of $n$ variables with truth table $\beta$. Let $h$ be a function mapping all nodes $v \in V$ to its relevant substring of $\beta$. Then $h(v)$ is a bead for all nodes $v \in V$.*

*Proof.* First note that the terminal nodes always correspond to the length 1 beads 0 and 1. We thus focus on internal nodes. Suppose an internal node $v$ is such that $h(v)$ is not a bead. Then its two children $v_0$ and $v_1$ found by following paths 0 and 1 respectively have equivalent representations, i.e. $h(v_0) = h(v_1)$. This means that the two strings $h(v_0)$ and $h(v_1)$ are the same length. The substrings being the same length implies that both nodes are of the same depth in a full tree representation of the BDD and hence are labelled with the same Boolean variable. Furthermore, having the same substring also means that the subtrees starting with $v_0$ and $v_1$ as the root are the same. Thus, nodes $v_0$ and $v_1$ are equivalent, contradicting the claim that the BDD is reduced. $\square$

The following proposition shows that conditionally redundant features correspond directly to non-beads in any BDD representation of a given decision rule.

**Proposition 2.** Let $G_x$ be a BDD representation of the decision rule $F^r_{\phi,D} : \mathcal{S}_\phi \times \mathcal{A} \to \{0,1\}$. Then any node in $G_x$ labelled with a conditionally redundant feature will correspond to a non-bead.

*Proof.* Let $\phi_1 = \{p_1, \ldots, p_n\}$ be the predicate set and a predicate $p_i$ labels nodes on the $ith$ depth of $G_x$. Recall that a feature $p_j$ is conditionally redundant iff for all $s \in \mathcal{S}_{\phi_1}$, $F^r_{\phi_1,D}(s, a) = F^r_{\phi_2,D}(s', a)$ where $\phi_2 = \phi_1 \setminus \{p_j\}$ and $s' = s_{1:j-1}s_{j+1:n}$.

Suppose that a predicate $p_j$ is conditionally redundant. Let $a \in \{0,1\}^{j-1}$ and $b \in \{0,1\}^{n-j}$. Since $p_j$ is conditionally redundant, we must have $F_{\phi_1}(a0b) = F_{\phi_2}(ab)$. Similarly, $F_{\phi_1}(a1b) = F_{\phi_2}(ab)$. Thus, $F_{\phi_1}(a0b) = F_{\phi_1}(a1b)$. Therefore any node labelled by $p_j$ has a truth table that repeats a binary string of length $2^{j-1}$ about its midpoint and is thus a non-bead. $\square$

Theorem 4 guarantees that a BDD reduction procedure will remove all non-beads and Proposition 2 guarantees that conditionally redundant features correspond to non-beads in any BDD representation of a decision rule. Thus, Theorem 4 and Proposition 2 provide the basis upon which BDD reduction can be utilised to remove conditionally redundant features. The proof of Theorem 2 follows directly from combining Proposition 2 and Theorem 4.

Whilst Theorem 2 shows that BDD reduction can remove conditionally redundant features, it is important to note that the definition of a conditionally redundant feature is dependent upon the ordering of the predicates. This is part of the motivation for performing RF-BDD.

# F Coding $MDP$ Sequences and a $Cost_M$ Counter-example

## F.1 Coding $MDP$ Sequences

To evaluate candidate state abstractions, the $\Phi$MDP approach ranks mappings based on the code length of the state-action-reward sequence. For this approach to be well formed, the underlying interaction sequence must be ergodic: the frequencies of every finite substring of the sequence converge asymptotically. Consider the set of states $\{s_{t_1}, \ldots, s_{t_k}\}$ transitioned to from a particular state $s$ and action $a$. This set of states will be an i.i.d. sequence generated according to $\theta^a_{s,s'} = \mathbb{P}(s_{t_i} = s'|s, a)$. The state sequence can then be considered a combination of i.i.d. sequences and the probability of the state sequence conditioned on actions is given by

$$\mathbb{P}(s_{1:n}|a_{1:n}, \theta) = \prod_{s,a} \prod_{j=t_1}^{t_k} \mathbb{P}(s_j|s, a) = \prod_{s,a} \prod_{j=t_1}^{t_k} \theta^a_{s,s_j} = \prod_{s,a} \prod_{s'} \left(\theta^a_{s,s'}\right)^{n^a_{s,s'}} \tag{8}$$

In practice, $\theta$ is unknown and we instead rely on estimates $\hat{\theta}$. If $\theta^a_{s,s'}$ is estimated by a frequency estimator $\hat{\theta}^a_{s,s'} = \frac{n^a_{s,s'}}{n^a_s}$, then the code length of $s_{1:n}$ given $a_{1:n}$ is given by

$$CL(s_{1:n}|a_{1:n}) = -\log \mathbb{P}(s_{1:n}|a_{1:n}, \hat{\theta}) = -\sum_{s,a} \sum_{s'} n^a_{s,s'} \log(\hat{\theta}^a_{s,s'}) = \sum_{s,a} CL(\hat{\boldsymbol{\theta}}^{\boldsymbol{a}}_{\boldsymbol{s}}), \tag{9}$$

where $\hat{\boldsymbol{\theta}}^{\boldsymbol{a}}_{\boldsymbol{s}} = (\hat{\theta}^a_{s,s'})_{s'}$ and $CL(\hat{\boldsymbol{\theta}}^{\boldsymbol{a}}_{\boldsymbol{s}}) = -\sum_{s'} n^a_{s,s'} \log(\hat{\theta}^a_{s,s'}) = n^a_s H(\boldsymbol{n^a_s}/n^a_s)$. Similarly, the code length for the reward sequence is given by $CL(r_{1:n}|s_{1:n}, a_{1:n}) = \sum_{s,a,s'} CL(\hat{\boldsymbol{\theta}}^{\boldsymbol{a}}_{\boldsymbol{s,s'}})$, where $\hat{\boldsymbol{\theta}}^{\boldsymbol{a}}_{\boldsymbol{s,s'}} = (\hat{\theta}^{a,r}_{s,s'})_r$ and $CL(\hat{\boldsymbol{\theta}}^{\boldsymbol{a}}_{\boldsymbol{s,s'}}) := \sum_{s,a,s'} n^{a,r}_{s,s'} \log(\theta^{a,r}_{s,s'})$.

## F.2 $Cost_M$ Counter-example

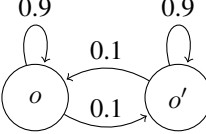

Figure 10: A diagram illustrating the observation transition probabilities of the environment in the counter-example.

Consider the counter-example presented in Figure 10. The environment emits two observations $o, o'$ and two rewards 0, 1 with the following distribution:

$$p(o_t|h_{t-1}, a_{t-1}) = \begin{cases} 0.9 & \text{if } o_t = o_{t-1} \\ 0.1 & \text{otherwise.} \end{cases}$$

if $o_t \neq o_{t-1}$:

$$p(r_t|h_{t-1}, a_{t-1}, o_t) = \begin{cases} 0.1 & \text{if } r_t = 1 \\ 0.9 & \text{if } r_t = 0 \end{cases}$$

if $o_t = o_{t-1}$:

$$p(r_t|h_{t-1}, a_{t-1}, o_t) = \begin{cases} 0 & \text{if } r_t = 1 \\ 1 & \text{if } r_t = 0 \end{cases}$$

Note that the given percept dynamics of the given environment are effectively independent of the actions performed. Now consider two candidate MDP mappings $\phi_0$ and $\phi_1$. For all $h_t \in \mathcal{H}$,

$$\phi_0(h_t) = \begin{cases} 1 & \text{if } o_{t-1} \neq o_t \\ 0 & \text{otherwise.} \end{cases}$$

$$\phi_1(h_t) = 0 \,.$$

Clearly the mapping $\phi_0$ is more informative and provides direct indication of the dynamics under which positive reward is received. Let us consider $Cost_M$ in the limit normalised by $n$. For ergodic sequences, the frequency estimates will converge to the true probabilities. Thus we have

$$\lim_{n \to \infty} \frac{Cost_M(n, \phi)}{n} = \lim_{n \to \infty} \frac{CL(s_{1:n}|a_{1:n}) + CL(r_{1:n}|s_{1:n}, a_{1:n}) + CL(\phi)}{n}$$

$$\overset{(a)}{=} -\lim_{n \to \infty} \sum_{s,a} \sum_{s'} \frac{n_{s,s'}^a}{n} \log\left(\frac{n_{s,s'}^a}{n_s^a}\right) - \lim_{n \to \infty} \sum_{s,a} \sum_{r} \frac{n_s^{ar}}{n} \log\left(\frac{n_s^{ar}}{n_s^a}\right)$$

$$\overset{(b)}{=} -\sum_{s,a} p_\phi(s,a) \sum_{s'} p_\phi(s'|s,a) \log p_\phi(s'|s,a) -$$

$$\sum_{s,a} p_\phi(s,a) \sum_{r} p_\phi(r|s,a) \log p_\phi(r|s,a)$$

$$\overset{(c)}{=} -\sum_s p_\phi(s) \sum_{s'} p_\phi(s'|s) \log(p_\phi(s'|s)) - \sum_s p_\phi(s) \sum_r p_\phi(r|s) \log p_\phi(r|s)$$

$$\overset{(d)}{=} \sum_s p_\phi(s) H_\phi(S'|S = s) + \sum_s p_\phi(s) H_\phi(R|S = s)$$

Here (a) follows by definition, (b) follows since the frequency estimates converge, (c) follows since the actions do not affect the state or reward probabilities and (d) uses the definition of conditional entropy where we subscript by $\phi$ to indicate that the distribution is under the $\phi$ mapping.

We can now compute $\lim_{n \to \infty} \frac{Cost_M(n, \phi_0)}{n}$:

$$\sum_s p_{\phi_0}(s) H_{\phi_0}(S'|S = s) = p_{\phi_0}(s = 0) H_{\phi_0}(S'|S = 0) + p_{\phi_0}(s = 1) H_{\phi_0}(S'|S = 1)$$

$$= 0.9 \cdot 0.47 + 0.1 \cdot 0.47$$
$$= 0.47$$

$$\sum_s p_{\phi_0}(s) H_{\phi_0}(R|S = s) = p_{\phi_0}(S = 0) H_{\phi_0}(R|S = 0) + p_{\phi_0}(S = 1) H_{\phi_0}(R|S = 1)$$

$$= 0.9 \cdot 0 + 0.1 \cdot 0.47$$
$$= 0.047$$

We thus have that $\lim_{n\to\infty} \frac{Cost_M(n,\phi_0)}{n} = 0.47 + 0.047 = 0.517$. For $\phi_1$ we have

$$\sum_s p_{\phi_1}(s) H_{\phi_1}(S'|S = s) = 0$$

$$\sum_s p_{\phi_1}(s) H_{\phi_1}(R|S = s) = p_{\phi_1}(S = 0) H_{\phi_1}(R|S = 0) + p_{\phi_1}(S = 1) H_{\phi_1}(R|S = 1)$$

$$= 0.08$$

We then have $\lim_{n\to\infty} \frac{Cost_M(n,\phi_1)}{n} = 0.08$. Thus the uninformative mapping $\phi_1$ minimises $Cost_M$ even though it is less predictive of the rewards. As mentioned earlier, this phenomenon occurs due to the heavy weighting given to the code length of the state sequence. Even if the model is no good for predicting reward, a simple transition model can cause decreases in the state sequence code length that can offset increases in the reward sequence code length. This issue will be exacerbated in environments where the reward signal is sparse; the cost associated with being unable to predict a sparse reward is negligible relative to the modest reduction in complexity of the state transition model.