# OpenReview forum: "A Direct Approximation of AIXI Using Logical State Abstractions"
_NeurIPS.cc/2022/Conference — NeurIPS 2022 Accept_

### Official Review · Reviewer_4Mza · 2022-06-21

**Rating:** 5
**Confidence:** 2
**Soundness:** 2 fair
**Presentation:** 3 good
**Contribution:** 2 fair

**Summary:**

This paper considers an AIXI reinforcement learning approach that combines state abstraction, model learning, and planning. For state abstraction, a subset of predicates from a large candidate set is selected on the basis of their ability to predict rewards. For model learning in the abstraction, a binarized version of CTW is used. For planning, a version of UCT is used. Experiments are in an epidemic domain that features a large structured state space with partial observability and stochastic transitions.


**Questions:**

1. How exactly do you fit the decision rules F? I am guessing, based on the name “Random Forest”, that something like decision tree learning is used (before converting to BDD)? This should be stated clearly.
2. L261 remarks that the selected features and BCTW induce an MDP with a small state space. Is the UCT algorithm here anything other than running UCT in this abstract MDP? If there is a difference, what is it? If not, I think the entire planning approach could be explained in just this way, without necessarily even mentioning \rho UCT.

**Limitations:**

Societal impact is discussed in the conclusion.


**Strengths And Weaknesses:**

### Strengths
* The epidemic domain considered in the experiments appears challenging and interesting, and it’s obviously timely.
* The writing of the paper is crisp and clear at the sentence level.
* The paper builds nicely on an existing line of work, but should engage more with other lines of work.
* The limitations in the results section are stated clearly, especially around L331.

### Weaknesses
* The experimental results do not include any baseline comparisons. The domain does appear difficult, but baseline results would be necessary to get a sense of whether the proposed approach is on par with or better than alternative approaches. As the paper states, the results are not optimal, but other approaches may or may not fare similarly. Relational reinforcement learning / statistical relational learning methods would be good choices. But holding part of the proposed approach fixed and considering alternative implementations of the other parts would also be illuminating. For example, I am curious whether alternative feature selection strategies, alternative model learning algorithms, or alternative planning algorithms would perform better than the proposed methods.
* Using alpha=0 in CostM (L120) arguably defeats the purpose of the CostM objective. If it were straightforward to extend the feature selection method to handle alpha>0, then using alpha=0 for simplicity in certain domains may be okay, but I am guessing that the extension is not straightforward. Presentation aside, I worry about using alpha=0 because in many domains, I would expect that certain features are not directly predictive of reward at the current time step, but are very important for predicting reward at a later time step.
* The treatment of reward as a categorical variable is limiting, but I know there are many works that do this. But then further converting the multiclass problem into multiple binary classification problems, as implied by the feature selection approach, seems even more limiting, because it really loses sight of the fact that rewards have a canonical ordering, and that there can only be one reward given at any time step.
* I appreciate the effort to make the paper self-contained, but I think the overall presentation would be more clear if the paper spent less time on background and more time focusing on novel contributions. For example, the BDD and UCT sections can probably be reduced significantly.
* L133 refers to “the practical convenience of working with (the functional subset of) a language like Python.” I am wondering, then, why Section 2.4 is important: why not simply state that predicates are enumerated from a domain-specific language (DSL), as in the program synthesis literature? In other words, for the sake of the rest of this paper, why do we need to constrain predicates to be anything other than functions from histories to booleans?
* The paper claims “the richest, direct approximation of AIXI known to date.” I wonder, though, if many existing approaches in the RL literature could be seen to approximate AIXI, even if they are not usually described through the AIXI lens. I suspect that this is the case, and in my opinion, this diminishes the significance of the work here, because there are many related works with variously rich representations that just use different framings.

### Minor
* The acronym BCTW is used on L139 ad L143 before it is defined on L212
* Typo: “abstractions that contains” on L149
* The second summation in Definition 5 has r where it should have r’
* The use of \mathcal{A} in Definition 7, for arbitrary finite alphabet, is confusing because that symbol was previously used for the action space
* The results refer to a “random selection of different action sequence indicator functions.” Does “random” here mean “seemingly arbitrary”? If so, that word choice or something else may be more clear.
* The font sizes in the plots in Figure 1 could be increased for clarity

---

> ### Author Response · Authors · 2022-08-02
> **Thank you for the review!**
>
> Thank you for your insightful and thoughtful comments. Our response to your questions/queries are as follows.
>
> * The experimental results do not include any baseline comparisons.
>
> We have included comparisons on the epidemic control problem in section C.2 in the supplementary material. Please also see Section C.2 for a description of the chosen baseline methods. The chosen methods are quite similar to relational reinforcement learning methods (e.g. Q-RRL in [4]) with the addition of iterative state abstractions. These two methods also present an alternative feature selection strategy, i.e. greedy expansion of the state based upon a heuristic, and an alternate model learning strategy, i.e. frequency estimate of transition model. Alternative planning methods were not considered as the main contribution within our work is in the model learning component and good planning results depend heavily upon the ability to learn a good model.
>
> * alpha=0 in CostM (L120) arguably defeats the purpose of the CostM objective.
>
> Considering $CL(s_{1:n} | a_{1:n})$ for optimisation can lead to two key issues. The first issue is that the wrong solution can be favoured by the optimisation criteria. As the example in Section E.2 in the supplementary material shows, one can construe counter-examples where the value of $\alpha$ leads to the wrong state abstraction being favoured by the optimisation objective. A second issue is that since $CL(s_{1:n} | a_{1:n})$ is optimised independent of any reward signal, it does not inherently provide any information about the long-term reward. Thus, it is best thought of as a regularisation term that can make the resulting MDP easier to learn.
>
> * I worry about using alpha=0... certain features are not directly predictive of reward at the current time step, but are very important for predicting reward at a later time step.
>
> The point raised here is an important one and highlights a subtle distinction in our setup. The perspective put forward here represents the 'forward' view where features at the current time step can be important for predicting reward at a future time step. Our framework instead allows long-range dependencies to be captured through the 'backward' view: the reward at the current time step is allowed to depend arbitrarily far back into the history sequence. This history dependency can be captured by the predicate functions. In the backward view, predicting instantaneous rewards is sufficient.
> For example, consider an n-Markov environment. In the forward view, the reward n steps into the future can depend (not only) on the current state. In the backward view, the reward at the current time step can depend on the state n steps into the past. If predicates are defined that capture a sufficient statistic of the past n steps, then the n-step dependency will be captured.
>
> * ...rewards have a canonical ordering, and that there can only be one reward given at any time step
>
> The canonical ordering and the fact that only one reward is given at any time step is not crucial given the framing of the $\Phi$ MDP model learning problem. The $Cost_M$ objective is really an optimization objective focused on minimizing the entropy of the conditional distribution of rewards given state-actions (see Section E.1 in supplementary). Working with distributions means the canonical ordering of rewards is not important. A crucial and standard assumption for the $\Phi$ MDP approach is that the resulting state-reward-action sequence is ergodic. This assumption ensures the distributions are guaranteed to be well-formed given enough data and also mean that the time factor is not important. In this context, the binary classification formulation is not as limiting as initially considered. The binary classes are defined to detect a property of the underlying reward distributions, namely whether they are D-sharp. Whilst there are limitations to this heuristic approach as it does not directly optimize the $Cost_M$ objective (which is left to $\Phi$ - BCTW once features are selected), the limitations are not those considered in the original comment.
>
> * why not simply state that predicates are enumerated from a domain-specific language (DSL), as in the program synthesis literature?
>
> We acknowledge that DSLs can be used when it is advantageous to do so. However, it is not always the case that a DSL is preferred over a general purpose functional logic programming language.
>
> [4] Saso Dzeroski, Luc De Raedt, and Hendrik Blockeel, _Relational reinforcement learning._ ICML, 1998.

---

> > ### Author Response · Authors · 2022-08-02
> > **Rebuttal continued...**
> >
> > * The paper claims “the richest, direct approximation of AIXI known to date.” ...
> >
> > Whilst many existing approaches in the RL literature could be seen to approximate AIXI, our claim that our approach is the 'richest' and most 'direct' approximation is supported by two important factors. Firstly, one of the key components of AIXI is that the environment model is computed as an exact Bayesian mixture model over possible environments. The $\Phi$ - BCTW method computes an exact mixture model. This is not something typically considered by other model-based approaches and is a crucial argument for why we claim a 'direct' approximation to AIXI. Secondly, existing approaches that compute a mixture model are typically constrained to a particular model. For example, Bayesian RL methods [5] typically consider a fixed parametrised model and perform mixtures over parameters. The expressivity of using predicate functions on histories defined in higher-order logic instead allows a much larger hypothesis set and does not constrain the mixture model to a fixed parametrised model. Hence, we claim that our approach is the richest approximation to AIXI to date.
> >
> > * How exactly do you fit the decision rules F?
> >
> > The decision rules are fit by randomly sampling subsets of features before simply using frequency estimates to record the conditional reward distribution. We will make this more clear in the final manuscript.
> >
> > * Is the UCT algorithm here anything other than running UCT in this abstract MDP? If there is a difference, what is it?
> >
> > The UCT algorithm is standard. We will shrink the exposition on UCT in the final manuscript.
> >
> > The other minor comments will also be amended in the final manuscript.
> >
> > [5] Mohammed Ghavamzadeh, Shie Mannor, Joelle Pineau, Aviv Tammar. _Bayesian Reinforcement Learning: A Survey_. Retrieved from https://arxiv.org/abs/1609.04436

---

> > > ### Comment · Reviewer_4Mza · 2022-08-04
> > > **Thanks for the responses**
> > >
> > > Thanks for the thorough response!
> > >
> > > **New Baselines**
> > > - In Appendix C.1, it's stated that "the top thirty highest weighting features were chosen" for the random forest feature selection baseline. Can you clarify what "highest weighting" means here? Also, why not use the same feature selection method with the same threshold that's used by RF-BDD?
> > > - The new baselines and domains in Appendices C.2-3 seem to significantly strengthen the paper.
> > >
> > > **On $\alpha = 0$**
> > > - The backward view helps me understand, thanks for clarifying. I would recommend mentioning something to this effect in the paper.
> > > - I also think that the current presentation of the objective is a bit circuitous. An objective is defined (Equation 4), but it's not the ideal objective, so half of it gets thrown away by setting $\alpha = 0$. It would be clearer for me if it the paper just stated the actual objective first. The connection to the full CL may then be interesting to some, but it could be discussed later or in a footnote.
> > >
> > > **Reward ordering and discretization**
> > > - This is a lesser point, but I am not sure if I fully understood the response. To clarify, my original critique was that in reality, rewards are real numbers. Utility theory, the MDP formalism, etc. rely heavily on this. The model presented in this work pretends that rewards are not real numbers for the sake of developing a tractable optimization objective. My feeling is that this loss of information about the real structure of rewards should be avoidable.
> > >
> > > **Domain-specific language**
> > > - To say that this work uses a general purpose functional logic programming language is misleading. The features are extremely domain-specific (_NaiveInfectionRate_, _InfectionRateOfChange_, etc.).
> > > - I still think that this subsection of the paper could be stated more simply, and think that the current writing distracts from the main contributions.
> > >
> > > **AIXI framing**
> > > - I take your point here to some degree, but also stand by my original comment.
> > >
> > > Given the new baselines and experiments and the other clarifications acknowledged above, I will raise my recommendation.

---

> > > > ### Author Response · Authors · 2022-08-09
> > > > **Response continued...**
> > > >
> > > > Thank you for your additional comments and newly considered recommendation. In regards to your remaining questions:
> > > >
> > > > *  Can you clarify what "highest weighting" means here? Also, why not use the same feature selection method with the same threshold that's used by RF-BDD?
> > > >
> > > > Random Forest provides a variable importance measure and that was used to determine the weighting of a feature. The same feature selection method was not used as Random Forest was run with rewards treated as a categorical target.
> > > >
> > > > * Reward ordering and discretization
> > > >
> > > > While feature selection is done by discretising the reward, both the CTW model learning and MCTS planning use rewards (i.e. real numbers) as they are without discretisation. If the underlying environment has a finite set of possible rewards, the order of the real-valued rewards is not that important when learning how to predict the instantaneous reward, which is what we do in the feature selection step. However, the order of the real-valued rewards is important when computing Q values, and MCTS and CTW both process rewards in their natural real-number form.

---

### Official Review · Reviewer_2MNv · 2022-06-30

**Rating:** 6
**Confidence:** 3
**Soundness:** 2 fair
**Presentation:** 2 fair
**Contribution:** 2 fair

**Summary:**

    This paper proposes a new approximation of AIXI which extends the previous work "A Monte-Carlo AIXI approximation". This novel approximation supports complex features on a non-Markovian and structured environment using formal logic. The main contributions of this paper can be listed as follow: (1) In order to select the right feature from the complex candidate features, use the BDD to remove redundant features. (2) Inspired by random forests, this paper proposed the RF-BDD algorithm to select the subset right feature using the D-sharp as a decision rule.
    Finally, The Experiment result shows that $\Phi$-AIXI-CTW agent model can handle the non-trivial contact network. It demonstrates that this agent can learn different behaviors under different environment configurations. In the feature selection process, the RF-BDD approach obtain a better result compared to the Random Forest feature selection.


**Questions:**

1. In general, what configuration of RF-BDD will get a great informative feature result?
2. Dose RF-BDD significantly increase the time cost compared to random forest feature selection?


**Limitations:**

Yes, author mentioned that current agent does not carry out privacy constraints, and the follow-up needs to continue to study the performance under the constraints of privacy constraints. Make it possible to actually apply.

**Strengths And Weaknesses:**

Originality:
	This work is a novel combination of well-known techniques to extend previous work, specifically, abstracting the logic state border the approximation of AIXI, using $\Phi$MDP as feature selection criterion, running BCTW for the mixture environment model, and getting the policy by $\rho$UCT.
	Some new methods are also introduced in this paper. In the feature selection, this paper proposes MDP-mapping, i.e., D-sharp, and then converts the feature selection to a classification task. Finally, this paper proposes the RF-BDD to do the feature selection, which makes it possible to tackle the large classes of problems.

Quality:
	The submission is somehow technically sound. The experimental results demonstrate the effectiveness of the method. The following aspects of the paper are still insufficient.
	1. The RF-BDD only provides enough information in the epidemic model. What about scaling ability and quality in other baselines?
	2. In feature selection comparison, this paper only makes the comparison with Random Forest feature selection. What about other methods?
Clarity:
	I think some of the points in the paper are not clear enough and not well organized.
	1. In section 2.3, we use the $\Phi$MDP as a feature selection criterion, however, in section 3.1, you actually use the D-sharp heuristic to do the feature selection. I don't know the relationship between them. Is it reasonable to replace it? Does it drive the feature selection result in a small $Cost_M$?
	2. The organization of the paper needs to be optimized, when I read section 4.3, the paper discusses Experiments 3 and 4, but I can't quickly locate their figures.
	3. In the experiment, I can't find details about BDDs constructed by RF-BDD to do the feature selection. How many times do you randomly select features to construct BDD for feature selection? In each time, how do you determine the subset of feature predicates to run the BDD reduction?

Significant:
	This paper proposes the approximation of AIXI over the non-Markovian domain and structured knowledge by logical state abstractions, which broadens the approximation of AIXI agent model. This paper provides a new idea to approximation and feature selection via RF-BDD.

---

> ### Author Response · Authors · 2022-08-02
> **Thank you for the review!**
>
> Thank you for your insightful and thoughtful comments. Our response to your questions/queries are as follows.
>
> * The RF-BDD only provides enough information in the epidemic model. What about...quality in other baselines?
>
> We provide additional experiments in 4 domains (see section C.3 in supplementary material) to demonstrate the efficacy of RF-BDD in other environments. We also provide comparisons on the original epidemic control domain (Section C.2 in supplementary). For more details on the baseline comparison methods, please see Section C.2 . As can be seen in Figure 6, our proposed method either outperforms the baselines or performs on par in these simpler domains and RF-BDD seems to provide enough information to perform well.
>
> * RF-BDD scaling ability, experimental configuration and time complexity.
>
> In our experiments, we perform BDD reduction on 500 different trees generated using a random sample of 8 features. The value of $D$ is adjusted to select a feasible amount of predicates for $\Phi$ - BCTW to deal with. We found this sufficient to achieve an informative result when considering predicate sets of size roughly 1500 on the epidemic problem and believe that larger sets of predicates can also be tackled with this configuration. This particular setup was also sufficient for the additional domains we tested. Smarter approaches to sampling features may reduce the number of samples required and is something we are looking into. In terms of time complexity, RF-BDD is bottlenecked by the time complexity of performing BDD reduction. BDD reduction is linear in the number of nodes in a tree. This can make RF-BDD slower than Random Forest feature selection for larger tree depths. However, the depth that can be feasibly used in practice is typically not very large as overfitting becomes an issue.
>
> * In feature selection comparison, this paper only makes the comparison with Random Forest feature selection. What about other methods?
>
> Some other feature selection methods were attempted. We were not able to tune Lasso to provide reasonable feature selection results and it always set too many features to zero. Forward selection methods (as in [3]) were also attempted but provided no performance advantage over Random Forest and also exhibited the potential for issues with variance. Figure 7 demonstrates some of the variability of using forward selection when compared with Random Forest in the Jackpot environment. Finally, our baseline comparisons can be viewed as alternative feature selection methods that perform the task of selecting features for state abstraction iteratively in an online fashion.
>
> * Relationship between D-sharp and $\Phi$ MDP. Is it reasonable to replace it? Does it drive the feature selection result in a small $Cost_M$?
>
> The $D$-sharp criteria does not directly optimise the $Cost_M$ criteria. Instead it provides a constraint on the set of possible solutions before $\Phi$ - BCTW optimises the $Cost_M$ objective over the given set of solutions. Note that the $Cost_M$ objective can be viewed as minimizing the entropy of possible solutions. The $D$-sharp criteria is an inductive bias that selects for solutions that allow each reward to be predictable and constrains the entropy of possible solutions; the larger $D$ is, the smaller the set of satisfying solutions. Finally, $\Phi$ - BCTW optimises the $Cost_M$ objective over the given set of solutions due to its fast convergence properties. We will make this more clear in the final manuscript.
>
> * Paper Organization
>
> The organization of the paper will be re-arranged and optimized for clarity in the final manuscript.
>
>
> [3] Gavin Brown, Adam Pocock, Ming-Jie Zhao, and Mikel Luján. _Conditional likelihood maximisiation: A unifying framework for information theoretic feature selection._ Journal of Machine Learning Research, 2012.

---

### Official Review · Reviewer_g5bg · 2022-07-10

**Rating:** 6
**Confidence:** 1
**Soundness:** 3 good
**Presentation:** 3 good
**Contribution:** 3 good

**Summary:**

This paper studies the problem of state abstraction in the general reinforcment learning setting. Specifically, the paper looks at AIXI agents, and addresses the challenge of feature selection for forming the state abstractions. In the experiments, the proposed model is evaluated on a timely control task of epidemic processes.
Please note that I am not knowledgeable in this research area and hence please lower the weight of my review.

**Questions:**

- I like the design and the results demonstrated in the experiments. The only concern I have is the lack of direct comparison to baseline methods, such as heuristic ones (Line 304), prior works on AIXI, or general RL methods.
- Apart from the considered control of epidemic processes, can the proposed method also apply to other complex sequential decision making problems?

**Limitations:**

The authors sufficiently discussed the potential limitations in the conclusion. The empirical study shows their method can be potentially a positive and timely contribution. I do not see any negative societal impact.

**Strengths And Weaknesses:**

- The considered problem of incorporating logical state abstraction in the AIXI framework is interesting and well motivated
- The problem formulation and the theoretical contributions are well explained. However, I do not have sufficient expertise to judge the correctness of the theoretical contributions.
- The experiment design is timely, and prevalant to the COVID-19 situation.

---

> ### Author Response · Authors · 2022-08-02
> **Thank you for the review!**
>
> Thank you for your comments. Our response to your questions are as follows.
>
> As it was a common criticism from all reviewers, we have provided experiments with baseline methods for comparison. We have chosen to compare against two decision-tree based, iterative state abstraction methods using splitting criteria as defined in U-Tree [1] and PARSS-DT [2]. In U-Tree, an existing node (representing a state) is split if splitting results in a statistically significant difference in the resulting Q-values as computed by a Kolmogorov-Smirnov test. In PARSS-DT, nodes are split if the resulting Q and V values are sufficiently far apart. Model learning is simply done by frequency estimation. To ensure our method does not have an informational advantage, both baseline methods are given the same initial set of predicate functions to consider node splits from.
>
> 1. In the updated supplementary material, we provide comparisons to our proposed method on the epidemic control problem. Figure 5 plots the learning curves between our proposed method and the two baseline methods on Experiments 3 and 4. Due to time constraints, the two baseline methods were only able to be run for roughly 500k steps. As can be seen, our proposed method outperforms the two baseline methods. The two baseline methods do not seem to be able to capture an informative state space to perform.
>
> 2. Our method can be applied to other complex sequential decision making problems as it is a general agent design framework. In the supplementary we provide experiments (with comparisons) on four domains: biased rock paper scissors (RPS), 2x5 Taxi, Jackpot, Stop Heist. For more details on each of the different domains, please refer to section C.3 in the supplementary material. Figure 6 shows that our proposed method either outperforms the baselines or performs on par. Whilst these domains are not complex, they do demonstrate the advantages of incorporating higher order models of computation as well as the generality of our approach and how there is no fundamental impediment to running on alternative domains.
>
> [1] Andrew Kachites McCallum. _Reinforcement Learning with Selective Perception and Hidden State._ PhD thesis, University of Rochester, 1996.
>
> [2] Jesse Hostetler, Alan Fern, and Thomas Dietterich. _Sample-based tree search with fixed and adaptive state abstractions._ JAIR, 2017.

---

> > ### Comment · Reviewer_g5bg · 2022-08-08
> > **Thanks for the response**
> >
> > Thank the authors for providing baseline performance to the epidemic experiments, as well as additional domains!

---

### Meta-Review · Area_Chair_Xee5 · 2022-08-23

**Recommendation:** Accept
**Confidence:** Less certain

**Metareview:**

The paper proposes an interesting approach and analysis for extending the AIXI agent to handle non-Markovian structured decision processes. Technically, the paper appears to be fine, and its evaluation has significantly improved after the addition of baselines. At the same time, this work could be strengthened by motivating its own significance more explicitly: despite being conceptually appealing, the practical importance of extending the AIXI agent to non-Markovian and structured environments is not entirely obvious.

**Award:**

No

---

### Decision · Program_Chairs · 2022-09-14

Accept